# Stimulation of endogenous cardioblasts by exogenous cell therapy after myocardial infarction

Konstantinos Malliaras, Ahmed Ibrahim, Eleni Tseliou, Weixin Liu, Baiming Sun, Ryan C Middleton, Jeffrey Seinfeld, Lai Wang, Behrooz G Sharifi & Eduardo Marbán*

## Abstract

Controversy surrounds the identity, origin, and physiologic role of endogenous cardiomyocyte progenitors in adult mammals. Using an inducible genetic labeling approach to identify small non-myocyte cells expressing cardiac markers, we find that activated endogenous cardioblasts are rarely evident in the normal adult mouse heart. However, myocardial infarction results in significant cardioblast activation at the site of injury. Genetically labeled isolated cardioblasts express cardiac transcription factors and sarcomeric proteins, exhibit spontaneous contractions, and form mature cardiomyocytes *in vivo* after injection into unlabeled recipient hearts. The activated cardioblasts do not arise from hematogenous seeding, cardiomyocyte dedifferentiation, or mere expansion of a preformed progenitor pool. Cell therapy with cardiosphere-derived cells amplifies innate cardioblast-mediated tissue regeneration, in part through the secretion of stromal cell-derived factor 1 by transplanted cells. Thus, stimulation of endogenous cardioblasts by exogenous cells mediates therapeutic regeneration of injured myocardium.

**Keywords** cardiac stem cells; cardioblasts; cardiomyocyte progenitor cells; cardiosphere-derived cells; heart regeneration
**Subject Categories** Cardiovascular System; Stem Cells

## Introduction

The adult mammalian heart is now viewed as a dynamic organ, capable of endogenous regeneration. While the rate of cardiomyocyte renewal in adults is low (Bergmann *et al*, 2009), exploitation of innate regenerative processes may open up novel therapeutic options favoring cardiomyocyte repopulation in the diseased heart. Improved insights into the cellular and molecular mechanisms of endogenous heart regeneration are needed, as the field is rife with controversy (Laflamme & Murry, 2011; Steinhauser & Lee, 2011; Garbern & Lee, 2013). While adult cardiomyocytes retain a detectable

(albeit small) ability to proliferate (Soonpaa & Field, 1997; Malliaras *et al*, 2013a; Senyo *et al*, 2013), the role of endogenous progenitor cells in adult cardiomyogenesis is unclear (Laflamme & Murry, 2011; Steinhauser & Lee, 2011; Garbern & Lee, 2013; Koudstaal *et al*, 2013). During the past decade, numerous populations of putative adult endogenous cardiomyocyte progenitors have been proposed, mainly on the basis of expression of surface markers and functional properties that have been used to mark progenitors in other organs, and expression of cardiac proteins *in vitro* or after delivery into recipient hearts following *ex vivo* expansion (Beltrami *et al*, 2003; Oh *et al*, 2003; Martin *et al*, 2004; Messina *et al*, 2004; Ott *et al*, 2007; Chong *et al*, 2011; Tamura *et al*, 2011). However, the physiologic importance of these various progenitor populations and their contribution to cardiomyocyte replenishment in the normal or injured adult heart remain topics of intense debate.

Genetic fate mapping offers the unique ability to study organ regeneration *in vivo* (Kretzschmar & Watt, 2012). Using an inducible fate mapping approach [where Cre recombinase activity, driven by the cardiac α-myosin heavy chain (αMHC) promoter, is induced prior to myocardial infarction (MI) to genetically label pre-existing cardiomyocytes], multiple groups have detected a dilution of the labeled myocyte pool post-injury (Hsieh *et al*, 2007; Loffredo *et al*, 2011; Qian *et al*, 2012; Song *et al*, 2012), consistent with the generation of new cardiomyocytes by unlabeled myocyte progenitors. By combining the aforementioned experimental approach (which retrospectively captures the net result of endogenous progenitor cell activation and differentiation, but fails to reveal the identity and properties of said progenitors) with long-term BrdU pulsing, we have previously provided indirect evidence that cell therapy with exogenous cardiosphere-derived cells (CDCs) upregulates endogenous progenitor cell-mediated heart regeneration (Malliaras *et al*, 2013a). Here, we took a different approach: induction of Cre recombinase activity following (and not prior to) myocardial injury. We sought to achieve prospective genetic labeling of endogenous cardioblasts, as they turn on the αMHC promoter post-injury on their way to cardiomyogenic differentiation. Using this approach, we directly investigated the existence, identity, and origin of cardioblasts in the injured adult mouse heart, and how these progenitors are influenced by exogenous cell therapy.

Cedars-Sinai Heart Institute, Los Angeles, CA, USA
*Corresponding author. Tel: +1 310 423 7557; Fax: +1 310 423 7637; E-mail: Eduardo.Marban@csmc.edu

# Results

## Activation of endogenous cardioblasts post-MI

In bitransgenic Myh6-MerCreMer/ZEG mice, transient exposure to tamoxifen induces an irreversible genetic switch from β-galactosidase expression to green fluorescent protein (GFP) expression in cells with αMHC promoter activity, resulting in permanent labeling of αMHC$^+$ cells by GFP (Hsieh et al, 2007). Bitransgenic mice were randomized to undergo sham surgery or MI by permanent ligation of the left anterior coronary artery (LAD), followed by daily pulsing with 4OH-tamoxifen (Fig 1A). Ten days later, harsh enzymatic digestion of the explanted hearts (followed by multiple filtering steps) resulted in efficient depletion of adult cardiomyocytes (Supplementary Fig S1). Examination of the isolated cardiomyocyte-depleted cell fraction revealed small (11.5 ± 3.7 μm) round GFP$^+$, lacZ$^-$ cells which we go on to implicate as cardioblasts (Fig 1D, E and I). Quantification by flow cytometry and epifluorescence microscopy revealed that GFP$^+$ cardioblasts were rare in sham-operated hearts (0.12 ± 0.06% of non-myocyte cells) (Fig 1B and C, Supplementary Fig S2), but their number increased > 10-fold post-MI (1.34 ± 0.48% of non-myocyte cells in the risk area) (Fig 1B and C). GFP$^+$ cardioblasts were highly enriched in the infarct and peri-infarct area (Fig 1F, Supplementary Fig S3), compared to the remote myocardium (Fig 1G and H). These results demonstrate that MI results in cardioblast activation (which includes 'turning on' the αMHC promoter) within the risk area.

## Properties of endogenous cardioblasts

We investigated expression of cardiac transcription and structural factors at the protein level in fluorescence-activated cell sorting (FACS)-sorted GFP$^+$ cardioblasts by fluorescent immunocytochemistry (without any intermediate culture step) and tissue immunohistochemistry. The majority of GFP$^+$ cardioblasts expressed NKX2-5 (69%) and GATA4 (74%), while 16% were positive for MEF2C (Fig 2B and D, Supplementary Fig S4). We could not detect expression of TBX5 or Isl1 in GFP$^+$ cardioblasts. With regard to sarcomeric proteins, 38% of GFP$^+$ cardioblasts expressed α-sarcomeric actinin and 39% expressed αMHC (Fig 2C and D). The discrepancy between αMHC promoter activity and protein expression (αMHC at the protein level was expressed in only a subset of GFP$^+$ cardioblasts) can be rationalized by the fact that promoter activity is several steps upstream of protein synthesis, one of several factors underlying the poor correlation between mRNA and corresponding cellular protein abundance (Vogel & Marcotte, 2012). No GFP$^+$ cardioblasts expressed endothelial (CD31) or smooth muscle cell (α-smooth muscle actin) markers. Profiling of proteins associated with cell cycle progression revealed that 17% of GFP$^+$ cardioblasts were positive for Ki67 (a marker of late G1, S, G2, and M phases of cell cycle) and 2% were positive for phosphorylated histone H3 (H3P, a marker of karyokinesis) (Fig 2E and G, Supplementary Fig S5A). Immunohistochemistry identified GFP$^+$ cardioblasts appearing to undergo mitosis in the border zone (Fig 2F and Supplementary Fig S5B).

To investigate whether GFP$^+$ cardioblasts exhibit contractile activity, enzymatically digested cardiomyocyte-depleted heart cells were plated onto fibronectin-coated dishes (without FACS for GFP,

to avoid traumatic cell injury). While few GFP$^+$ cardioblasts survived in culture, approximately 11% (7/66) exhibited weak spontaneous contractions in vitro over the course of the first 2 days post-plating, without exposure to cardiac differentiation medium (Fig 2A, Supplementary Movies S1 and S2). Modifications in the culture conditions (including plating isolated cardiac cell preparations onto a feeder layer, or culturing cells in embryonic stem cell medium) failed to improve survival of GFP$^+$ cardioblasts in vitro. While rare examples of dividing GFP$^+$ cardioblasts could be detected (Supplementary Fig S6), no GFP$^+$ cardioblasts survived more than 1 week in culture.

To investigate expression of cardiogenic molecules and cardiac structural factors at the transcript level, we performed RT–qPCR in RNA extracted from FACS-sorted GFP$^+$ and GFP$^-$ myocyte-depleted cardiac cells (obtained from the same hearts, n = 5 hearts) and from adult cardiomyocytes (n = 3) for genes that are upregulated during cardiomyogenic differentiation of embryonic stem cells (Paige et al, 2012). With regard to cardiac transcription factors and cardiogenic molecules, we found upregulation of NKX2-5 (7.8-fold), MEF2C (3.8-fold), TBX3 (2.6-fold), FOXC2 (5.1-fold), and BMP2 (5.4-fold) expression in GFP$^+$ cardioblasts compared to GFP$^-$ cardiac cells. A non-significant increase in GATA4 expression (1.8-fold, P = 0.151 versus GFP$^-$ cells) was observed [a finding that may also reflect GATA4 expression in GFP$^-$ cardiac fibroblasts (Zaglia et al, 2009)], while no major differences in expression of ISL1, TBX2, TBX5, TBX20, MESP1, MESP2, GATA6, WNT2, WNT5a, WNT11, DKK1, HAND1, HAND2, FOXC1, and SMARCD3 could be detected. Compared to adult cardiomyocytes, GFP$^+$ cardioblasts exhibited lower expression of NKX2-5, similar expression of MEF2C, TBX3, and GATA4, and increased expression of BMP2 and FOXC2 (Fig 3). With regard to cardiac structural factors, GFP$^+$ cardioblasts exhibited an 'intermediate' phenotype between GFP$^-$ non-myocyte cardiac cells and adult cardiomyocytes. Specifically, we observed upregulation of MYH6 (13.5-fold), MYH7 (7.7-fold), MYL2 (15.9-fold), MYL3 (13.7-fold), TNNI3 (5.5-fold), TNNT2 (6.7-fold), MYBPC3 (16.4-fold), PLN (2.4-fold), DES (4.0-fold), NPPA (4.0-fold), CKM (11.0-fold), and TTN (3.8-fold) in GFP$^+$ cardioblasts compared to GFP$^-$ cardiac cells. Compared to adult cardiomyocytes, GFP$^+$ cardioblasts exhibited decreased expression of all investigated cardiac structural factors, with the exception of MYH7 (which was expressed at similar levels in both cell populations). Importantly, expression of fibroblast (VIM)-, endothelial (VWF)-, and smooth muscle cell (ACTA2)-related genes was lower in GFP$^+$ cardioblasts compared to GFP$^-$ non-myocyte cardiac cells, providing additional evidence that FACS for GFP$^+$ myocyte-depleted cardiac cells enriches a committed cardioblast population (Fig 3).

To examine whether GFP$^+$ cardioblasts expressed surface markers or exhibited functional properties previously attributed to various putative endogenous cardiomyocyte progenitor populations, we looked for expression of Sca-1 (Oh et al, 2003), c-kit (Beltrami et al, 2003), nestin (Tamura et al, 2011), PDGFRa (Chong et al, 2011), and stage-specific embryonic antigen 1 (SSEA1) (Ott et al, 2007) in GFP$^+$ cardioblasts, as well as the ability to actively efflux Hoechst dye (side population cells) (Martin et al, 2004) by immunocytochemistry and flow cytometry. We found that 54.5 ± 15.3% of GFP$^+$ cardioblasts (as determined by immunocytochemistry) were positive for Sca-1, but Sca-1 was not specific for cardioblasts (only ~3% of Sca1$^+$ cells expressed GFP; Fig 4A and D). A minority of

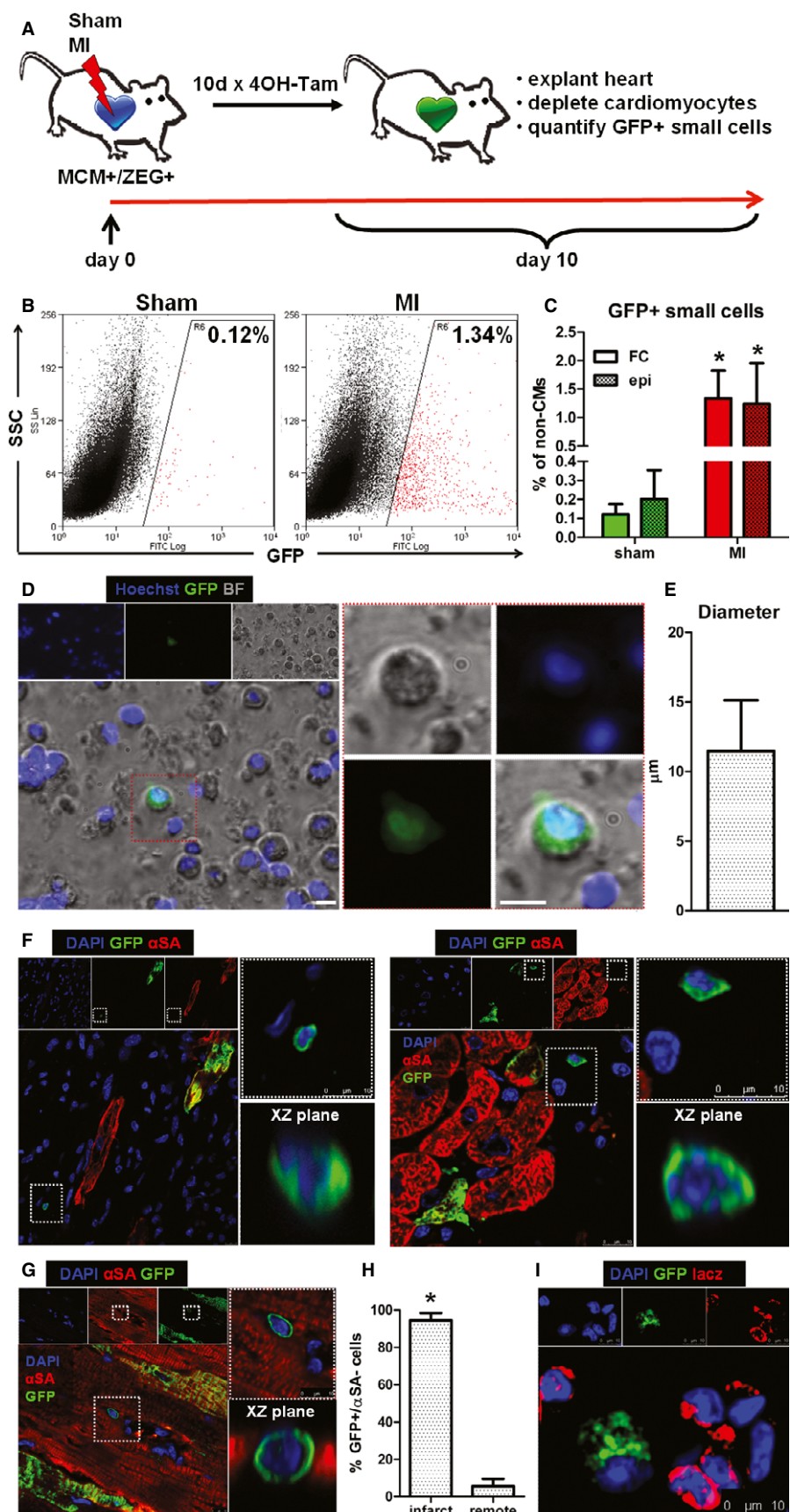

Figure 1.

GFP$^+$ cardioblasts (10.8 ± 3.2%, as determined by immunocytochemistry) expressed nestin (Fig 4C and D), while < 1% of GFP$^+$ cardioblasts actively effluxed Hoechst dye (Fig 4B and D). Immunocytochemistry demonstrated that GFP$^+$ cardioblasts were negative for c-kit, SSEA1, PDGFRα, and CD34 (Fig 4D). To exclude the possibility that enzymatic tissue digestion altered the expression of surface markers via cleavage of stem cell-related antigens, we performed RT–qPCR in RNA extracted from FACS-sorted GFP$^+$ myocyte-depleted cardiac cells (obtained from five hearts) for expression of KIT, FUT4 (SSEA1), PDGFRa, and CD34. KIT and FUT4 (SSEA1) were not detected in any samples. Very low expression of PDGRFα and CD34 was detected in 3/5 samples [on average > 1,000-fold less compared to expression of MYH6 (αMHC) in the same samples] (Supplementary Fig S7). However, we cannot exclude the possibility that expression of stem cell-related antigens may be lost as progenitors begin to differentiate.

To investigate whether GFP$^+$ cardioblasts can differentiate into mature cardiomyocytes *in vivo*, small cells isolated from infarcted mice 10 days following MI and daily tamoxifen pulsing were FACS-sorted for GFP positivity and lack of 7-actinomycin D incorporation (to select for live cells). We then injected these cardioblasts into normal (n = 3) or infarcted (n = 3) recipient hearts of background non-transgenic mice (10,000 cardioblasts/heart, Fig 5A). Employing a model of cardioblast isolation and transplantation into non-transgenic recipients was necessary, as our fate mapping model also labels pre-existing cardiomyocytes, complicating the investigation of cardiomyogenic differentiation of GFP$^+$ cardioblasts *in situ*. Three months post-injection, recipient hearts were examined for GFP$^+$ cells, either by immunohistochemistry or by epifluorescent microscopy of dissociated cardiomyocytes. Rare events of cardiomyogenic differentiation could be detected in four of six cardioblast-injected hearts [2/3 normal (4.2 ± 2.2 GFP$^+$ cardiomyocytes/heart section) and 2/3 infarcted (3.4 ± 3.6 GFP+ cardiomyocytes/heart section) hearts; Fig 5B–D]. GFP$^+$ cardiomyocytes appeared fully mature and structurally integrated with the surrounding myocardium, as they shared gap junctions with neighboring GFP$^-$ myocytes (Fig 5B). While we cannot exclude cell fusion (between injected GFP$^+$ cells and host myocytes) as a source of GFP$^+$ cardiomyocytes, such fusion events are exceedingly rare in the myocardium (Alvarez-Dolado et al, 2003), occurring at a frequency 10- to 100-fold smaller

compared to the frequency of GFP$^+$ cardiomyocytes observed post-injection of GFP$^+$ cardioblasts here. No instances of GFP$^+$ endothelial (von Willebrand factor$^+$) or smooth muscle cells (α-smooth muscle actin$^+$) cells could be detected. Even though we did not perform a controlled study to investigate whether transplantation of GFP$^+$ cardioblasts confers therapeutic benefit to infarcted mice, comparison of cardiac function by echocardiography between baseline and 5 weeks post-MI revealed a significant decline in ejection fraction (30 ± 5% versus 22 ± 7%, P = 0.036). The lack of major functional benefit can be rationalized by the paucity of long-term engrafted GFP$^+$ cardiomyocytes. The latter may reflect the low dose of injected GFP$^+$ cardioblasts, likely compounded by limited survival of GFP$^+$ cardioblasts following traumatic FACS purification and transplantation into recipient hearts; however, a low efficiency of mature cardiomyogenic differentiation by GFP$^+$ cardioblasts *in vivo* cannot be excluded.

### Origin of endogenous cardioblasts

Since contribution of bone marrow-derived cells to cardiomyogenesis is controversial in the adult mammalian heart (Laflamme et al, 2002; Quaini et al, 2002; Deb et al, 2003; Balsam et al, 2004; Murry et al, 2004; Nygren et al, 2008), we sought to investigate whether activated GFP$^+$ cardioblasts arise from hematogenous seeding. Immunohistochemistry of infarcted hearts demonstrated that GFP$^+$ cardioblasts were negative for the pan-leukocyte marker CD45 (Supplementary Fig S8). However, since bone marrow also contains CD45$^-$ cells that have been reported to express cardiac proteins (Kucia et al, 2006; Dawn et al, 2008), we performed bone marrow transplantation experiments in order to investigate definitively a potential bone marrow origin of GFP$^+$ cardioblasts. Bone marrow was isolated from bitransgenic Myh6-MerCreMer mice and transplanted into lethally irradiated non-transgenic background mice (n = 9) (Wang et al, 2006). Four weeks later (a time period sufficient to ensure bone marrow reconstitution by donor cells), transplanted mice were randomized to undergo sham surgery (n = 4) or MI (n = 5), followed by daily pulsing with 4OH-tamoxifen (Fig 6A). Ten days later, lacZ$^+$ cells were readily detectable in the infarct region (Fig 6B), indicating successful reconstitution of the bone marrow by transplanted cells obtained from bitransgenic

**Figure 1.  Endogenous cardioblasts are activated post-myocardial infarction.**

A    Study schematic. Bitransgenic Myh6-MerCreMer/ZEG mice were used, in which transient exposure to 4OH-tamoxifen induces an irreversible genetic switch from β-galactosidase expression to green fluorescent protein (GFP) expression in cells with α-myosin heavy chain (αMHC) promoter activity, resulting in permanent labeling of αMHC$^+$ cells by GFP.

B    Representative flow cytometry plots of enzymatically digested myocyte-depleted cardiac cell preparations for side scatter (SSC) and GFP expression (color gating has been applied to the images). Numbers indicate average % GFP positivity in each group.

C    Pooled data for quantification of GFP$^+$ cardioblasts by flow cytometry (FC) and epifluorescence microscopy (epi) in sham-operated (sham) and infarcted (MI) mouse hearts (*P < 0.05 compared to sham, n = 5 mice/group).

D    Fluorescence micrograph of a GFP$^+$ cardioblast in a freshly isolated enzymatically digested myocyte-depleted cardiac cell preparation from an infarcted heart. Red-bordered area is magnified on the right (scale bar, 10 μm) [blue: Hoechst, green: GFP, bright field (BF)].

E    Diameter of GFP$^+$ cardioblasts measured by immunocytochemistry (n = 30 cells).

F–H  Confocal microscopy of tissue sections from infarcted hearts revealed increased number of activated GFP$^+$ cardioblasts in the infarct (F). The infarcted area is identified by the lack of cardiomyocytes (negative for αSA) and the presence of non-myocyte (αSA$^-$/DAPI$^+$) cells. GFP$^+$ cardioblasts were rare in the remote myocardium (G, H) (*P < 0.05 compared to remote, n = 4 mice). Partial labeling of resident cardiomyocytes (which also express αMHC) is observed. Images on the right are magnified images of area marked on left. Images of confocal scanning across the XZ plane are also provided [blue: DAPI, green: GFP, red: α-sarcomeric actinin (αSA)].

I    Fluorescent immunocytochemistry revealed that GFP$^+$ cardioblasts were lacZ-negative, confirming the genetic switch from β-galactosidase expression to GFP expression (blue: DAPI, green GFP, red: lacZ).

     

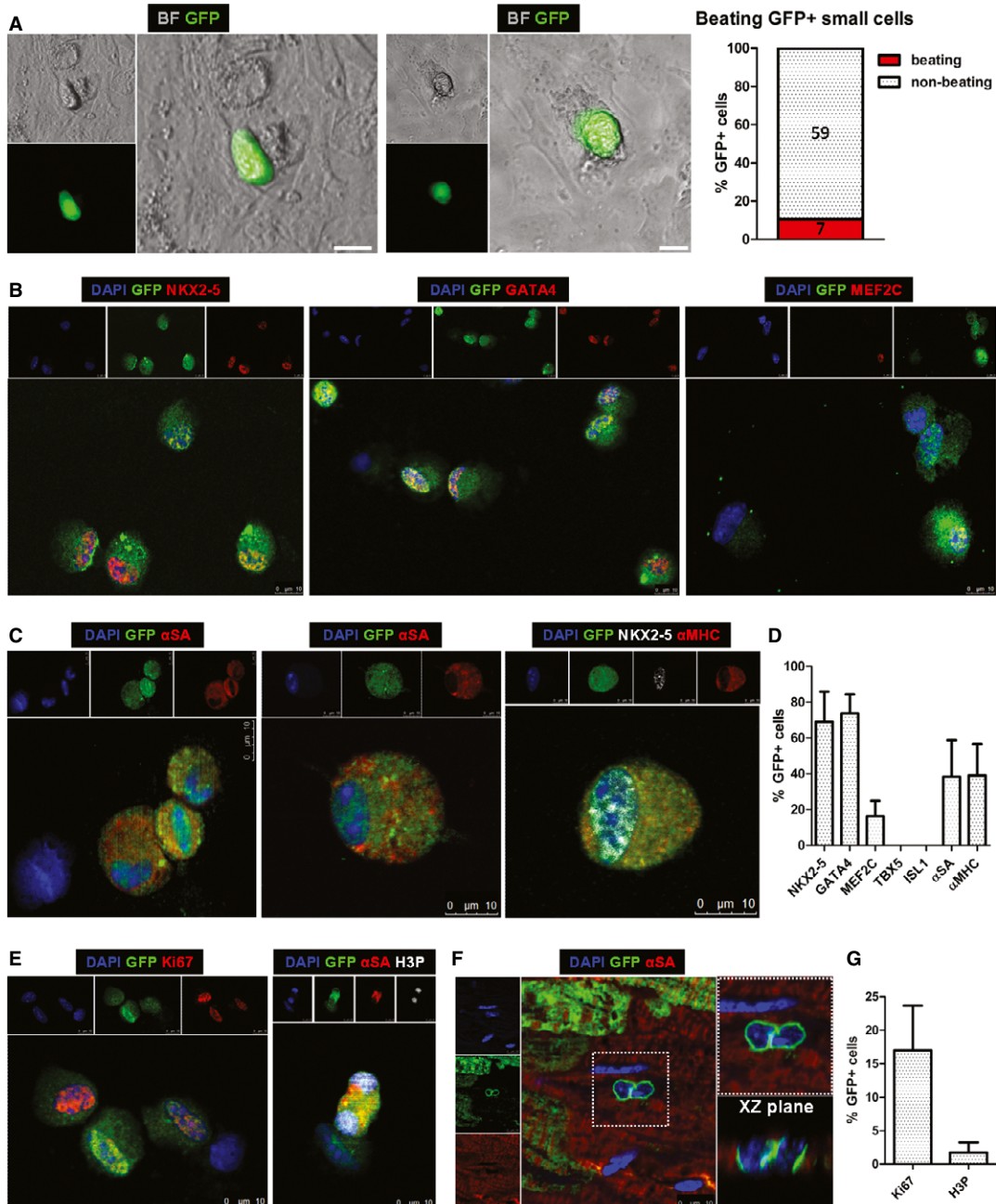

**Figure 2.  Endogenous cardioblasts exhibit spontaneous contractions and express cardiac transcription factors, sarcomeric and cycling proteins.**

A   Still images of GFP$^+$ cardioblasts exhibiting spontaneous contractile activity *in vitro* after 1 day in culture (see Supplementary Movies S1 and S2 for the corresponding Movies) (scale bars, 10 μm) [green: GFP, bright field (BF)].

B   Fluorescent immunocytochemistry of FACS-sorted GFP$^+$ cardioblasts for nuclear expression of cardiac transcription factors NKX2-5, GATA4, and MEF2C. Note that GFP$^−$ cells (middle & right panels, resulting from imperfect FACS for GFP) are negative for cardiac transcription factors (blue: DAPI, green: GFP, red: NKX2-5, GATA4, MEF2C).

C   Fluorescent immunocytochemistry of FACS-sorted GFP$^+$ cardioblasts for cytoplasmic expression of α-sarcomeric actinin (αSA) and α-myosin heavy chain (αMHC). Image on the right shows a GFP$^+$ cardioblast expressing αMHC in the cytoplasm and NKX2-5 in the nucleus. Note that GFP$^−$ cells (left panel) do not express sarcomeric proteins (blue: DAPI, green: GFP, red: αSA/αMHC).

D   Quantification of expression of cardiac transcription factors and sarcomeric proteins in GFP$^+$ cardioblasts by fluorescent immunocytochemistry (*n* = 4 mice). No expression of TBX5 or ISL1 could be detected in GFP$^+$ cells.

E   Fluorescent immunocytochemistry of FACS-sorted GFP$^+$ cardioblasts for Ki67 and phosphorylated histone H3 (H3P). Note that the GFP$^−$ cell is Ki67$^−$ (left panel) and the GFP$^+$ dividing cardioblast is expressing H3P in both nuclei (right panel) (blue: DAPI, green: GFP, red: Ki67/αSA, white: H3P).

F   Confocal microscopy revealed the presence of GFP$^+$ cardioblasts appearing to undergo mitosis in the border zone. Image on the right is a magnified image of highlighted area on left. Image of confocal scanning across the XZ plane is also provided (blue: DAPI, green: GFP, red: αSA).

G   Quantification of expression of cycling proteins in GFP$^+$ cardioblasts by fluorescent immunocytochemistry (*n* = 4 mice).

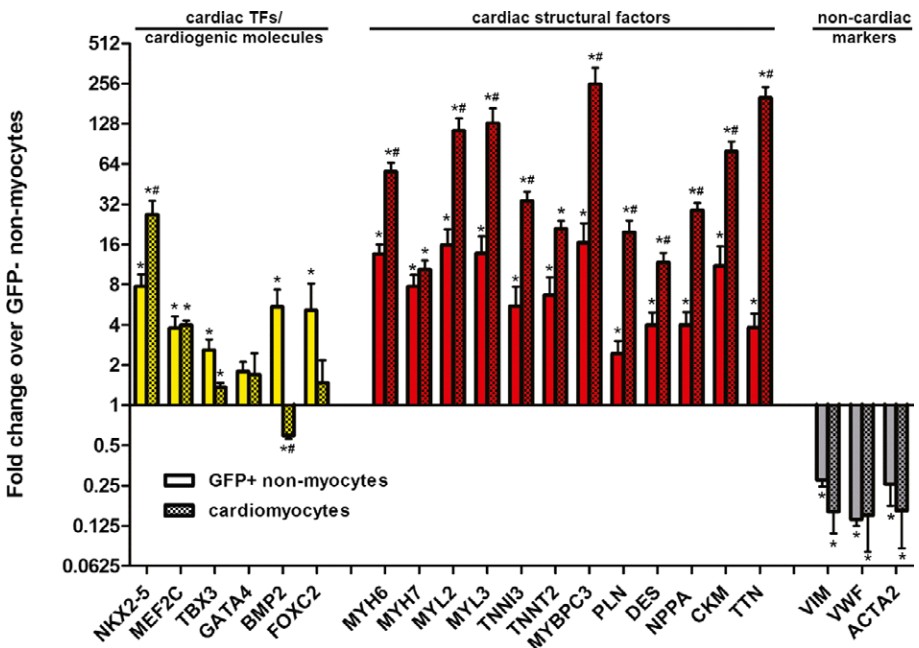

**Figure 3. Gene expression of cardiac transcription factors, cardiogenic molecules, and cardiac structural factors in endogenous cardioblasts.**
RT-qPCR in RNA extracted from FACS-sorted GFP$^+$ and GFP$^-$ myocyte-depleted cardiac cells revealed increased expression of cardiogenic molecules (NKX2-5, MEF2C, TBX3, FOXC2, and BMP2) and cardiac structural factors (MYH6, MYH7, MYL2, MYL3, TNNI3, TNNT2, MYBPC3, PLN, DES, NPPA, CKM, and TTN) in GFP$^+$ cardioblasts compared to GFP$^-$ cardiac cells. Compared to adult cardiomyocytes, GFP$^+$ cardioblasts exhibited lower expression of NKX2-5, and of all investigated cardiac structural factors (except of MYH7), similar expression of MEF2C, TBX3 and GATA4, and increased expression of BMP2 and FOXC2. Expression of fibroblast (VIM)-, endothelial (VWF)-, and smooth muscle cell (ACTA2)-related genes was downregulated in GFP$^+$ cardioblasts (*$P < 0.05$ compared to GFP$^-$ cells, #$P < 0.05$ compared to GFP$^+$ cells, $n$ = 5 mice for isolation of FACS-sorted GFP$^+$ and GFP$^-$ non-myocyte cells, $n$ = 3 mice for adult cardiomyocytes).

animals. Not a single GFP$^+$ cell could be detected by either tissue immunohistochemistry or epifluorescence microscopy and immuno-cytochemistry of enzymatically digested myocyte-depleted cell preparations isolated from sham-operated and infarcted hearts (Fig 6B and D). Flow cytometry revealed a percentage of GFP$^+$ cells similar to that measured in background non-transgenic (non-GFP-expressing) non-transplanted mice (~0.04% of cells were detected as GFP$^+$), which did not increase after MI (Fig 6C and D). These results exclude the possibility that GFP$^+$ cardioblasts arise from hematogenous seeding.

To investigate whether the increase in GFP$^+$ cardioblasts observed post-MI originates from dedifferentiation of resident myocytes or from expansion of a pre-existing pool of already committed ($\alpha$MHC$^+$) progenitors post-injury, non-infarcted bitransgenic mice underwent daily 4OH-tamoxifen pulsing for 10 days. Two weeks after completion of 4OH-tamoxifen pulsing [a time period sufficient to ensure complete 4OH-tamoxifen clearance, as 4OH-tamoxifen has a half-life of 6 h in the mouse (Robinson et al, 1991)], bitransgenic mice were randomized to undergo sham surgery or MI (Fig 6E). Ten days later, flow cytometry and epifluorescent microscopy revealed that the number of GFP$^+$ cardioblasts in pre-pulsed sham-operated hearts was similar to that measured in sham-operated hearts pulsed post-sham surgery [0.18 ± 0.11% (Fig 6F and G) versus 0.12 ± 0.06% (Fig 1B and C)]. While the number of GFP$^+$ cardioblasts increased in pre-pulsed hearts post-MI [0.44 ± 0.07% of cells in the risk area (Fig 6F and G)], it was lower than in infarcted hearts pulsed post-injury [1.34 ± 0.48% (Fig 1B and C)]. Thus, the majority of GFP$^+$ cardio-

blasts are activated (i.e., turn on the $\alpha$MHC promoter) post-MI, although a small minority may originate from expansion of a pre-existing, already committed cardioblast pool or from dedifferentiation of resident myocytes. While our model cannot differentiate between myocyte dedifferentiation and proliferation of pre-existing, already committed ($\alpha$MHC$^+$) progenitors (as both cell types are presumably labeled by 4OH-tamoxifen pulsing prior to injury), myocyte dedifferentiation appears the less likely possibility, as it has been shown to require more than 10 days in vitro (Zhang et al, 2010).

### SDF1-mediated upregulation of endogenous cardioblasts by cell therapy

Cardiosphere-derived cells (CDCs) (Smith et al, 2007) have been shown to exert regenerative effects in animal models of ischemic cardiomyopathy (Johnston et al, 2009; Malliaras et al, 2012, 2013b) and in patients with convalescent MI (Makkar et al, 2012; Malliaras et al, 2014). We previously concluded that CDCs upregulate endogenous progenitors, on the basis of conventional retrospective analysis (Malliaras et al, 2013a). Using the prospective genetic labeling approach described here, we directly tested whether transplantation of exogenous CDCs upregulates endogenous cardioblast activation in injured hearts. Bitransgenic mice were randomly assigned to 1 of 3 groups: (a) sham surgery, (b) MI and (c) MI followed by intramyo-cardial injection of mouse CDCs ($2 \times 10^5$, grown from background non-transgenic animals) into the infarct border zone. Mice were subsequently pulsed daily with 4OH-tamoxifen. Ten days later, flow

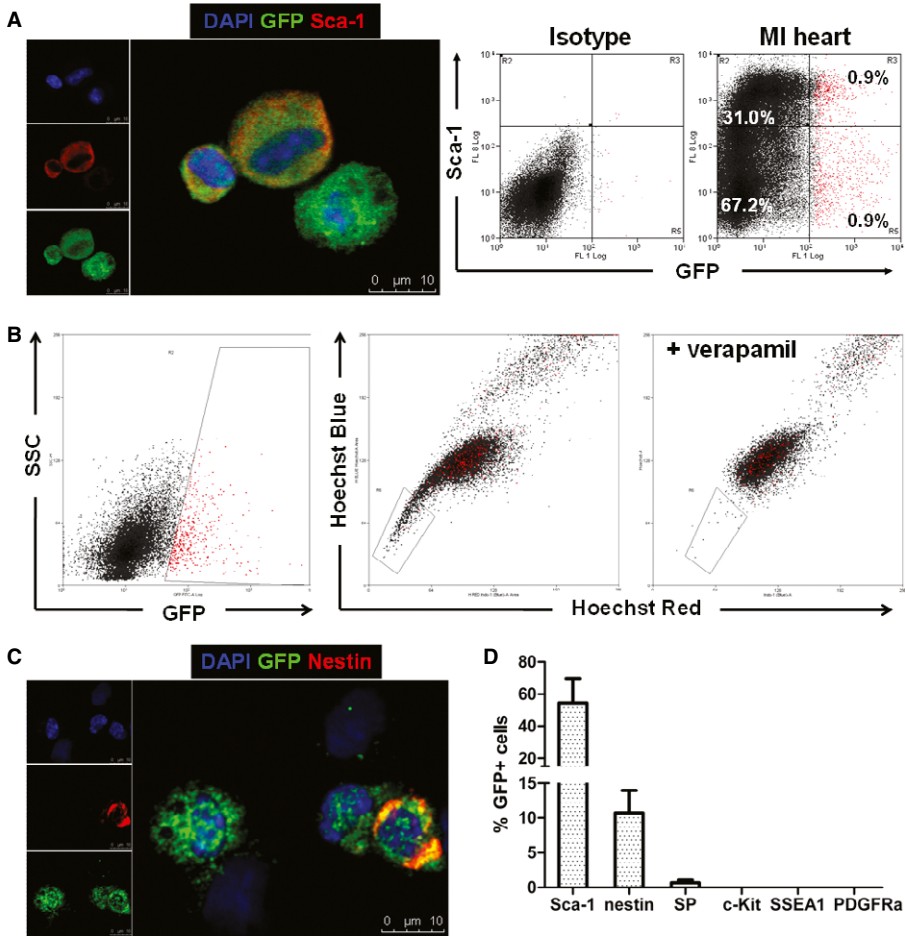

**Figure 4.  Expression of putative cardiac stem cell surface markers in endogenous cardioblasts.**

A   Fluorescent immunocytochemistry and flow cytometry for Sca-1 expression in GFP$^+$ cardioblasts (color gating has been applied to the flow cytometry plots) (blue: DAPI, green: GFP, red: Sca-1).

B   Flow cytometry demonstrated that < 1% of GFP$^+$ cardioblasts (which appear as red dots due to application of color gating) actively efflux Hoechst dye [side population (SP) cells]. The boxed area denotes the Hoechst-low SP cells. A sample co-incubated with verapamil and Hoechst served as the negative control, as verapamil abolishes the calcium-dependent Hoechst extrusion.

C   Fluorescent immunocytochemistry for nestin expression in GFP$^+$ endogenous cardioblasts (blue: DAPI, green: GFP, red: nestin).

D   Quantification of expression of surface markers previously employed to identify putative endogenous cardiac stem cells in GFP$^+$ cardioblasts by fluorescent immunocytochemistry (n = 3 mice). GFP$^+$ cardioblasts were uniformly negative for c-kit, SSEA1, and PDGFRα.

cytometry and epifluorescent microscopy revealed a significant upregulation of GFP$^+$ cardioblasts in infarcted hearts that had been injected with CDCs [4.78 ± 0.88% of non-myocyte cells in the risk area expressed GFP (Fig 7A)].

The indirect effects of CDCs to activate cardioblasts motivate a search for the relevant paracrine signals. Three sets of findings point to stromal cell-derived factor 1 (SDF1) and vascular endothelial growth factor (VEGF) as potentially important secreted factors in CDC-mediated stimulation of GFP$^+$ cardioblasts post-MI: (1) SDF1 (Segers *et al*, 2007; Unzek *et al*, 2007) and VEGF (Lin *et al*, 2012) can upregulate putative endogenous cardiomyocyte progenitors; (2) CDCs are potent secretors of both SDF1 and VEGF (Li *et al*, 2012); and (3) immunocytochemistry revealed that a subset of GFP$^+$ cardioblasts expressed CXCR4 (the receptor for SDF1, Fig 7B) and VEGFR1 (but not VEGFR2 or VEGFR3, Fig 7C). We therefore suppressed SDF1 or VEGF expression in CDCs by transduction with

short hairpin (sh)RNA-expressing lentiviral vectors. Knockdown of SDF1 and VEGF in CDCs decreased *in vitro* production of SDF1 [to levels ~18% of those produced by sh-control-transduced CDCs (Fig 7D)] and VEGF [~23% of the levels produced by sh-control-transduced CDCs (Fig 7E)] and decreased myocardial abundance of SDF1 (Fig 7F) and VEGF (Fig 7G) *in vivo*, 4 days after MI and injection of shRNA-transduced CDCs. Subsequently, bitransgenic mice underwent MI and were randomly assigned to receive intramyocardial injections of sh-control-transduced CDCs, sh-SDF1-transduced CDCs, or sh-VEGF-transduced CDCs, followed by daily pulsing with 4OH-tamoxifen. Ten days later, flow cytometry and epifluorescent microscopy revealed that knockdown of SDF1 (but not VEGF) in CDCs resulted in significant attenuation of CDC-mediated GFP$^+$ cardioblast upregulation in the infarcted area (Fig 7H).

We next sought to examine whether inhibition of SDF1-mediated stimulation of endogenous cardioblasts resulted in decreased

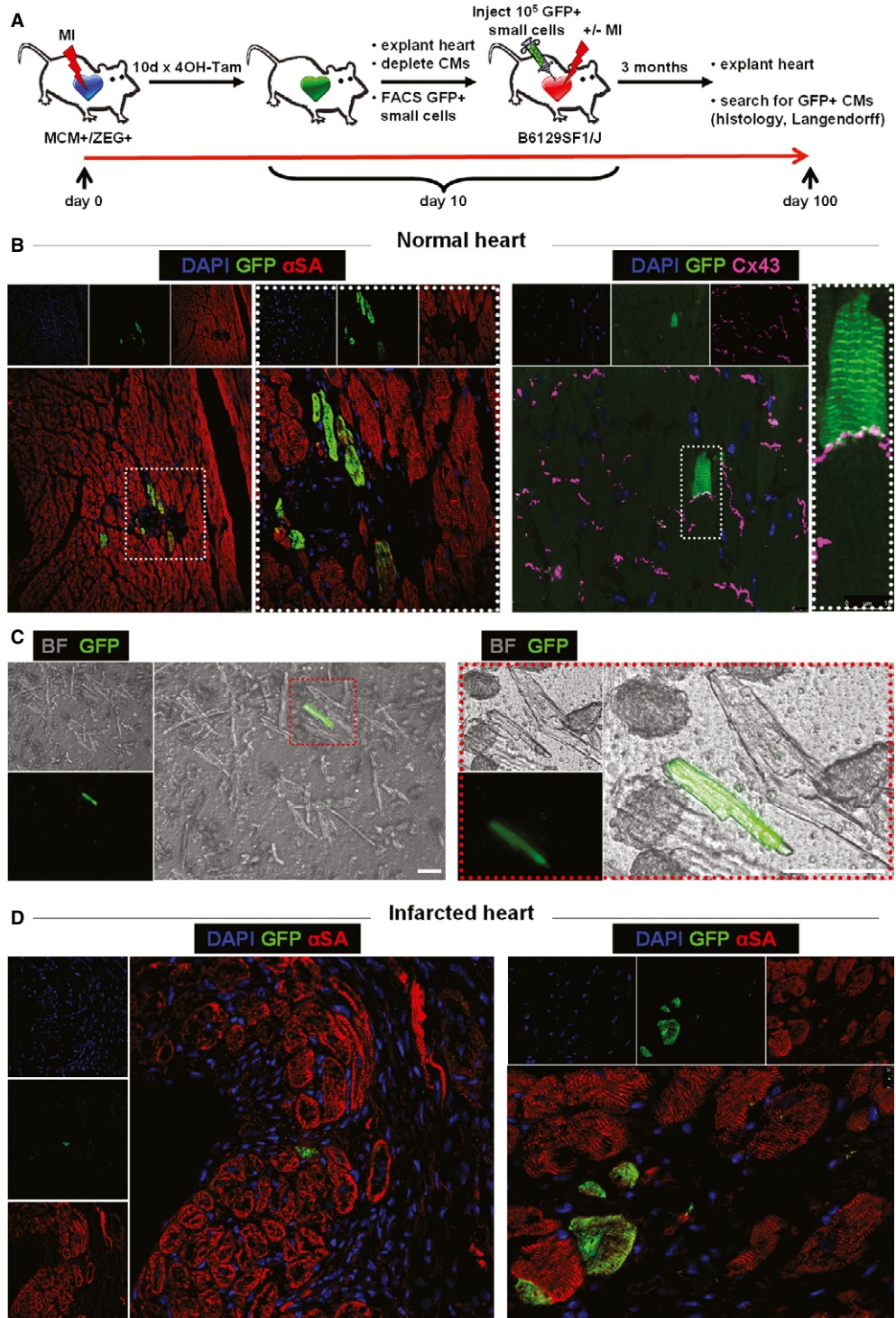

**Figure 5. Endogenous cardioblasts differentiate into mature myocytes after transplantation into recipient hearts.**

A   Study schematic. FACS-sorted GFP[+] cardioblasts activated post-MI were injected into non-injured (*n* = 3) or infarcted (*n* = 3) hearts of non-transgenic background mice.

B   Confocal microscopy in tissue sections from uninjured hearts revealed rare instances of GFP[+] cardiomyocytes 3 months post-injection. GFP[+] cardiomyocytes appeared fully mature and were connected by gap junctions [connexin 43 (Cx 43)] to neighboring GFP[−] myocytes. Images on the right are high-power images of highlighted areas on left [blue: DAPI, green: GFP, red: α-sarcomeric actin (αSA), purple: Cx 43].

C   Fluorescence micrograph of a GFP[+] cardiomyocyte after enzymatic dissociation of an uninjured heart on a Langendorff apparatus. Image on the right is a high-power image of inset on left (scale bars, 50 μm) [green: GFP, bright field (BF)].

D   Confocal microscopy of tissue sections from infarcted hearts revealed rare instances of GFP[+] cardiomyocytes in the infarct border zone 3 months post-injection (blue: DAPI, green: GFP, red: αSA).

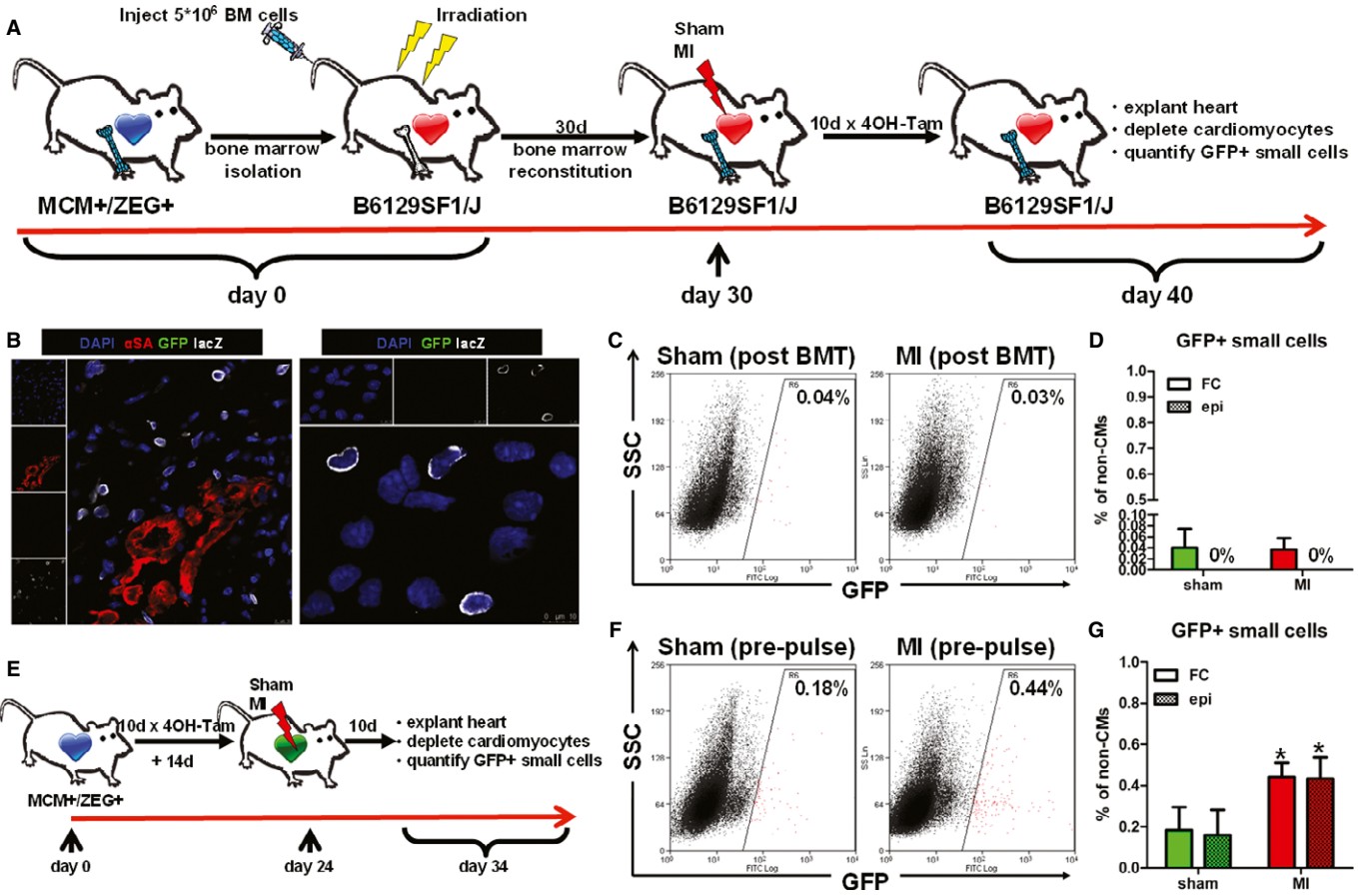

**Figure 6. Origins of endogenous cardioblasts.**

A Study schematic. Bone marrow from Myh6-MerCreMer was transplanted into lethally irradiated non-transgenic background mice. Bone marrow-reconstituted mice were randomized to undergo sham surgery or MI, followed by 4OH-tamoxifen pulsing.

B LacZ cells were readily detectable in the infarct region by immunohistochemistry (left) and immunocytochemistry (right), indicating successful reconstitution of the bone marrow (blue: DAPI, green: GFP, red: αSA, white: lacZ).

C Representative flow cytometry plots of enzymatically digested myocyte-depleted cardiac cell preparations for side scatter (SSC) and GFP expression (color gating has been applied to the images). Numbers indicate average % GFP positivity in each group.

D Quantification of GFP$^+$ cardioblasts by flow cytometry (FC) and epifluorescence microscopy (epi) in sham-operated (sham) and infarcted (MI) hearts from bone marrow-reconstituted mice. Not a single GFP$^+$ cell was detected by epifluorescence microscopy ($n = 4$–5 mice/group).

E Study schematic. Bitransgenic mice, pre-pulsed with 4OH-tamoxifen, were randomized to undergo sham surgery or MI.

F Representative flow cytometry plots of enzymatically digested myocyte-depleted cardiac cell preparations for side scatter (SSC) and GFP expression (color gating has been applied to the images). Numbers indicate average % GFP positivity in each group.

G Quantification of GFP$^+$ cardioblasts by flow cytometry (FC) and epifluorescence microscopy (epi) in sham-operated (sham) and infarcted (MI) hearts from pre-pulsed bitransgenic mice (*$P < 0.05$ compared to sham, $n = 3$–5 mice/group).

structural and functional benefits in the infarcted heart. Pre-pulsed bitransgenic mice were randomly assigned to undergo (a) MI, (b) MI followed by intramyocardial injection of sh-control-transduced CDCs, or (c) MI followed by intramyocardial injection of sh-SDF1-transduced CDCs. Mice pre-pulsed with 4OH-tamoxifen were used for this set of experiments in order to allow for comparison of the dilution of the pool of GFP$^+$ cardiomyocytes by endogenous progenitors (see below). Five weeks post-MI, both sh-control CDC- and sh-SDF1 CDC-treated hearts exhibited smaller scar mass and increased viable myocardium compared to untreated infarcted controls (Fig 8A–C). While no significant differences in total scar mass or total viable mass could be observed between sh-control CDC- and sh-SDF1 CDC-treated hearts, knockdown of SDF in transplanted CDCs increased scar transmurality and decreased viability in the

infarcted area (Fig 8A, D and E). Sh-control CDC-treated animals displayed superior global function, attenuation of adverse remodeling, and increased capillary density compared to sh-SDF1 CDC-treated animals 5 weeks post-MI, while both CDC-treated groups outperformed the infarcted controls (Fig 8F, Supplementary Figs S9 and S10). Quantification of the percentage of GFP$^+$ cardiomyocytes in the peri-infarct area by immunohistochemistry revealed a more pronounced dilution of the GFP$^+$ myocyte pool by GFP$^-$ cardiomyocytes in the sh-control CDC-treated hearts ($57.4 \pm 5.4\%$), compared to sh-SDF1 CDC-treated hearts ($66.1 \pm 4.8\%$) and infarcted control hearts ($73.5 \pm 4.6\%$)(Fig 8G). These results imply that the CDC-induced increase in replenishment of lost cardiomyocytes by endogenous progenitors post-MI is (at least partially) mediated through the secretion of SDF1 by the administered cells.

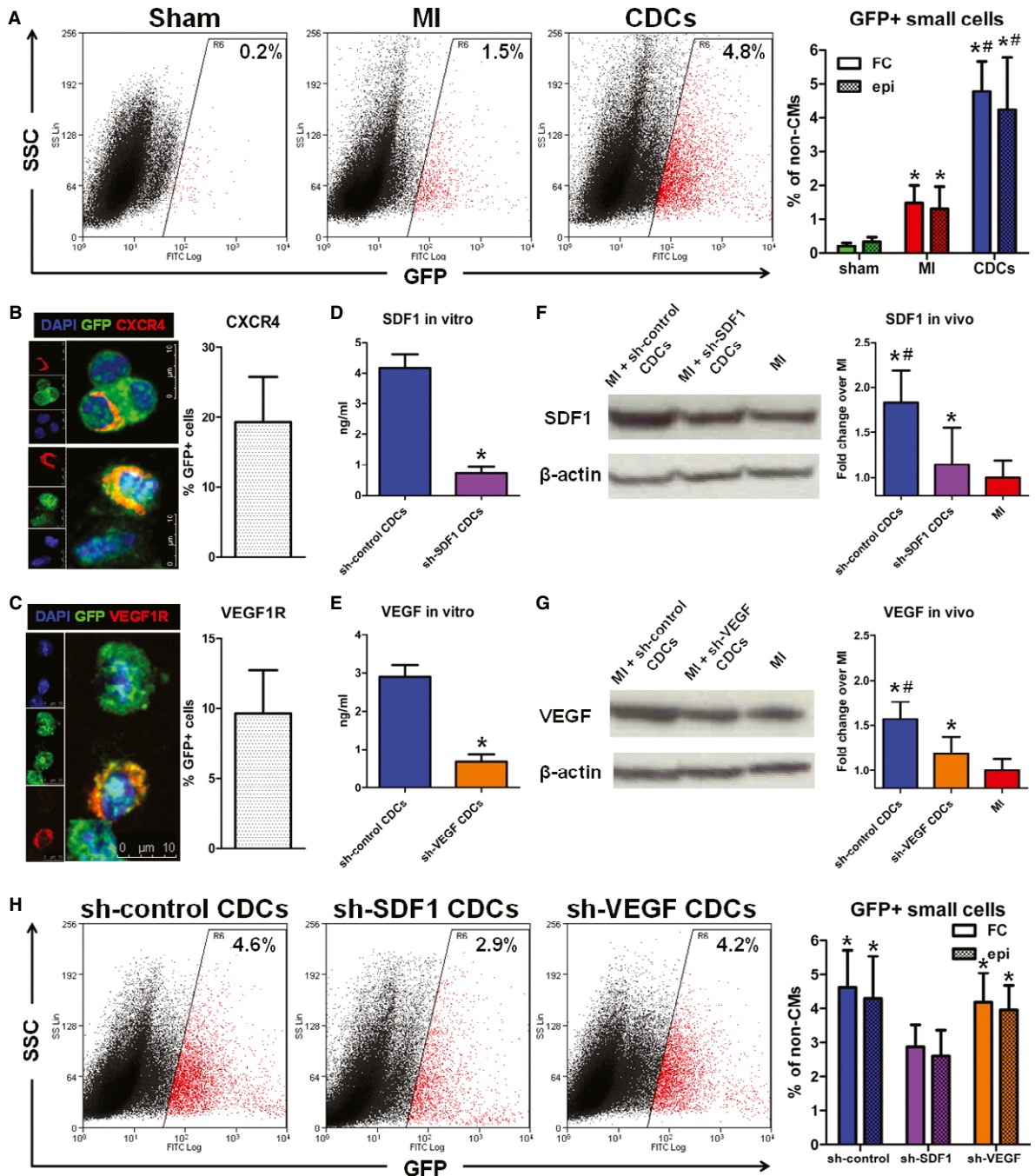

**Figure 7. SDF1-dependent cardiosphere-derived cell-mediated upregulation of endogenous cardioblasts post-MI.**

A     Representative flow cytometry plots of enzymatically digested myocyte-depleted cardiac cell preparations for side scatter (SSC) and GFP expression (color gating has been applied to the images). Numbers indicate average % GFP positivity in each group (left). Quantification of GFP+ cardioblasts by flow cytometry (FC) and epifluorescence microscopy (epi) in sham-operated (sham), infarcted (MI), and cell-treated (CDCs) mouse hearts (*P < 0.05 compared to sham, #P < 0.05 to MI, n = 3–5 mice/group).

B, C   Fluorescent immunocytochemistry and quantification of CXCR4 (B) and VEGFR1 (C) receptors in GFP+ cardioblasts (blue: DAPI, green: GFP, red: CXCR4/VEGFR1) (n = 3 mice).

D-G   Knockdown of SDF1 and VEGF resulted in decreased *in vitro* production of SDF1 (D) and VEGF (E) (*P < 0.05 compared to sh-control-transduced CDCs, n = 3) and in decreased myocardial levels of SDF1 (F) and VEGF (G) *in vivo*, 4 days after MI and injection of shRNA-transduced CDCs (*P < 0.05 compared to MI, #P < 0.05 compared to shSDF1- or shVEGF-transduced CDCs, n = 3 mice/group).

H     Representative flow cytometry plots of enzymatically digested myocyte-depleted cardiac cell preparations for side scatter (SSC) and GFP expression (color gating has been applied to the images). Numbers indicate average % GFP positivity in each group (left). Quantification of GFP+ cardioblasts by flow cytometry (FC) and epifluorescence microscopy (epi) in infarcted hearts injected with sh-control-, sh-SDF1- and sh-VEGF-transduced CDCs (*P < 0.05 compared to shSDF1-transduced CDCs, n = 5 mice/group).

## Discussion

Elucidation of the cellular sources of innate adult cardiomyogenesis is a crucial step toward the development of therapeutic strategies to amplify regeneration of infarcted myocardium. While adult cardiomyocytes can re-enter the cell cycle and proliferate post-MI (Soonpaa & Field, 1997; Malliaras et al, 2013a; Senyo et al, 2013), controversy surrounds the existence and identity of endogenous cardiomyocyte progenitors as well as their role in the generation of new myocytes in the injured adult mammalian heart. Previous studies using an inducible fate mapping approach (where Cre recombinase activity, driven by the αMHC promoter, is induced prior to myocardial injury) have provided indirect evidence that endogenous progenitors replenish lost cardiomyocytes post-MI (Hsieh et al, 2007; Loffredo et al, 2011; Qian et al, 2012; Song et al, 2012; Malliaras et al, 2013a). This approach cannot directly label said endogenous progenitors and thus cannot discern their molecular identity, origin, and functional properties. Expanding upon these studies, we used the same fate mapping model and hypothesized that induction of Cre recombinase activity after MI would enable prospective genetic labeling of endogenous progenitors, as they would presumably activate the αMHC promoter post-injury on their way to cardiomyogenic differentiation.

By employing the prospective labeling approach in combination with cardiac cell isolation protocols designed to achieve efficient myocyte depletion, we demonstrate the existence of rare non-myocyte cells exhibiting αMHC promoter activity in the adult uninjured heart. MI results in activation of the αMHC promoter in a substantial subset of non-myocyte cells in the risk area. These cells (which express cardiac transcription factors, cardiac structural proteins and proteins associated with cell cycle progression) can beat spontaneously in vitro and form mature, structurally integrated myocytes in vivo. Thus, the identified genetically labeled non-myocyte cells represent a physiologically relevant cell population with properties of endogenous cardioblasts. Our findings are in agreement with previous studies reporting activity of the αMHC promoter in myocyte progenitors (Zelarayan et al, 2008; Bailey et al, 2009; Dong et al, 2012; Hsueh et al, 2014), at least after they begin to differentiate. The identified GFP[+] cardioblasts exhibited spontaneous contractions early in culture without exposure to cardiomyogenic medium, in contrast to previously reported putative stem cell populations (Matsuura et al, 2004; Jesty et al, 2012); this difference may reflect the fact that our cells represent already committed, partially differentiated myocyte progenitors, rather than uncommitted, undifferentiated parent stem cells. With regard to their anatomic origin, we demonstrate that the identified endogenous cardioblasts do not arise from hematogenous seeding. While we show that the majority of cardioblasts are activated (i.e., turn on the αMHC promoter) post-MI, a small minority may originate from a pre-existing, already committed cardioblast pool or from dedifferentiation of resident myocytes.

Perhaps most importantly from a therapeutic and translational perspective, we find that cell therapy with CDCs stimulates endogenous cardioblast-mediated myocardial regeneration. CDCs have been shown to confer therapeutic benefit in animal models of acute MI and ischemic cardiomyopathy (Smith et al, 2007; Johnston et al, 2009; Malliaras et al, 2012, 2013b), and recently in humans (Makkar et al, 2012; Malliaras et al, 2014). In the prospective randomized CADUCEUS trial, intracoronary infusion of autologous CDCs in patients with convalescent MI induced an increase in viable myocardium of approximately 22 g (while no significant changes were observed in the control group) and resulted in improved regional contractility of the infarcted area (Makkar et al, 2012; Malliaras et al, 2014); these results are consistent with therapeutic heart regeneration. Multiple lines of evidence from preclinical studies suggest that the mechanism of benefit is indirect: CDCs stimulate endogenous reparative and regenerative pathways, resulting in durable benefits despite evanescent cell survival (Chimenti et al, 2010; Malliaras et al, 2013a). The indirect mechanism of action rationalizes the efficacy of allogeneic CDCs in animal models of MI (Malliaras et al, 2012, 2013b); indeed, allogeneic human CDCs have recently entered the clinical arena in the ongoing ALLSTAR trial (NCT01458405). The present study offers mechanistic insights into therapeutic regeneration: CDCs upregulate endogenous cardioblasts in part through the secretion of SDF1. Overexpression of SDF1 in mesenchymal stromal cells has been shown to increase their therapeutic efficacy (Zhang et al, 2007), while physical (Assmus et al, 2013)-, gene transfer (Penn et al, 2013)-, and protein (Segers et al, 2007)-based approaches that increase SDF1 myocardial levels have yielded promising results in animals and human subjects.

### Limitations

Our study has several limitations. First, our genetic labeling approach also labels resident cardiomyocytes. Since the presence of

---

**Figure 8.  Therapeutic stimulation of endogenous progenitor cell-mediated cardiac regeneration by cardiosphere-derived cells is at least partially dependent on SDF1 secretion.**

A    Representative images of Masson's trichrome-stained infarcted mouse hearts 5 weeks post-MI. Viable heart muscle stains red, while scar stains blue. Five consecutive sections per heart (obtained at 500-μm intervals, starting from the level of LAD ligation toward the apex) are presented. Images on the right are high-power images of the boxed areas on left, showing increased viable myocardium within the infarcted area in sh-control CDC-treated hearts, compared to sh-SDF1 CDC-treated hearts. The scar in infarcted controls (right) is largely transmural.

B–E   Morphometric analysis of hearts for the quantification of scar mass (B), viable mass (C), scar transmurality (D), and viability in the infarcted area (E) (*$P < 0.05$ compared to MI, #$P < 0.05$ compared to shSDF1-transduced CDCs, $n = 5$ mice/group).

F    Echocardiographic assessment of LV function (*$P < 0.05$ compared to MI, #$P < 0.05$ compared to shSDF1-transduced CDCs, $n = 5$ mice/group).

G    Representative fluorescent immunohistochemistry images of the border zone in sh-control CDC-treated hearts, sh-SDF1 CDC-treated hearts, infarcted control hearts, and sham-operated hearts (left). Quantification of the percentage of GFP[+] cardiomyocytes in the border zone (right) revealed a more pronounced dilution of the GFP[+] myocyte pool by GFP[−] cardiomyocytes in the sh-control CDC-treated hearts, compared to sh-SDF1 CDC-treated hearts and infarcted control hearts, suggesting increased replenishment of lost cardiomyocytes by endogenous progenitors (^$P < 0.05$ compared to sham, *$P < 0.05$ compared to MI, #$P < 0.05$ compared to shSDF1-transduced CDCs, $n = 5$–6 mice/group) (blue: DAPI, green: GFP, red: αSA).

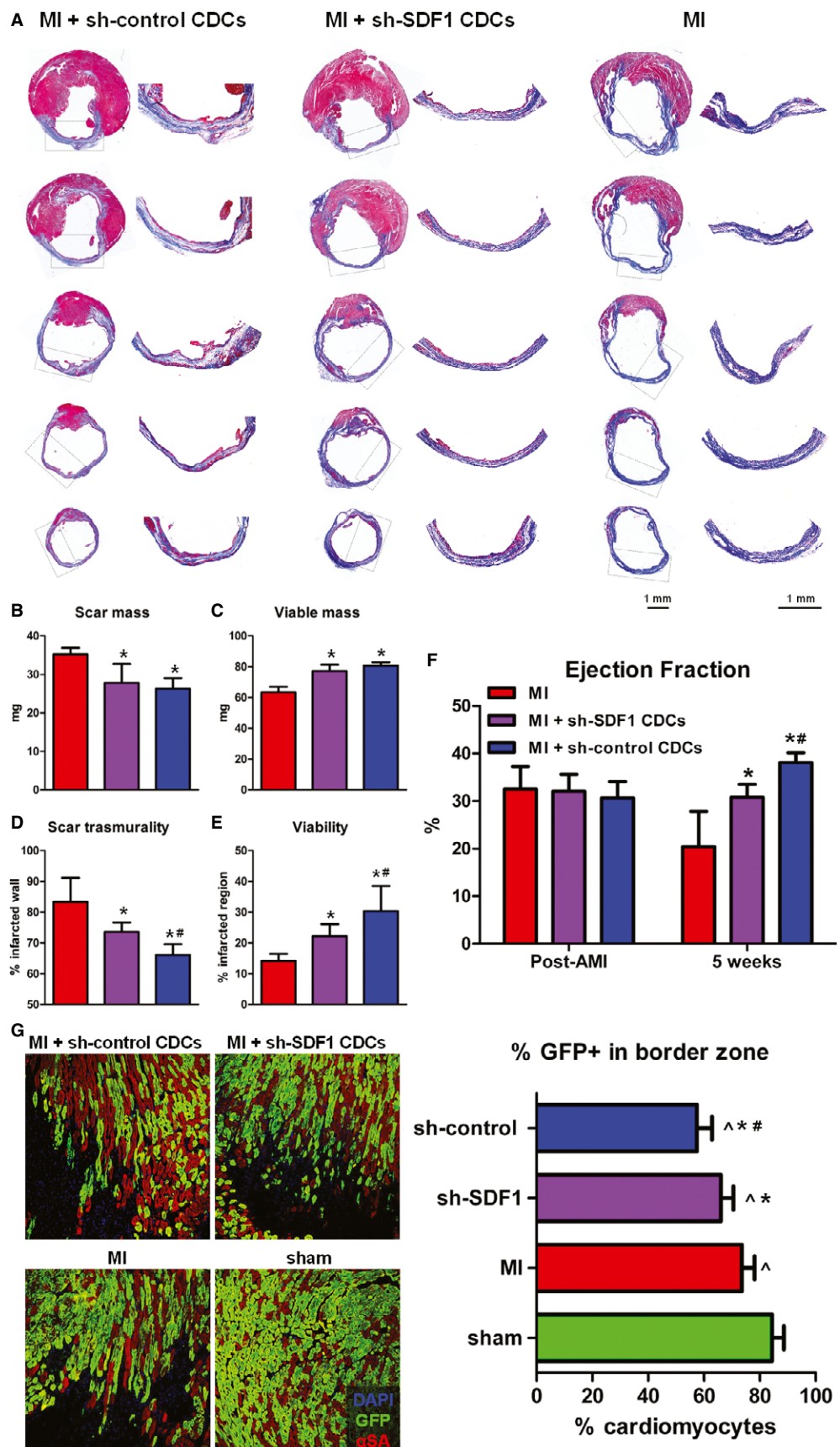

**Figure 8.**

labeled whole cardiomyocytes or cardiomyocyte remnants in the isolated cell preparations could confound our results, we took extensive care to ensure efficient cardiomyocyte depletion, by combining a harsh enzymatic digestion protocol with multiple filtering steps. We found that our approach effectively eliminated labeled resident cardiomyocytes from myocyte-depleted cardiac cell preparations (Supplementary Fig S1). In addition, in our analysis of cardiac cell preparations by epifluorescence microcopy and fluorescent immunocytochemistry, we only examined small cells [cell area in isolated GFP$^+$ cardioblasts was ~110 $\mu m^2$, much smaller compared to that of isolated neonatal (~330–870 $\mu m^2$) (Tamamori et al, 1998; Finn et al, 1999) or adult myocytes (~2500 $\mu m^2$) (Shioi et al, 2000)] with intact morphology and a central nucleus (to rule out myocyte fragments). In flow cytometry experiments, forward and side scatter gates were set to exclude cellular debris and cells with exceedingly high forward and side scatter [which are known characteristics of myocytes (Diez & Simm, 1998; Malliaras et al, 2013a)]. In tissue immunohistochemistry experiments, only cells positive for GFP but negative for sarcomeric proteins were considered as cardioblasts. In any case, if our cell isolation approach resulted in a significant contamination of cell preparations by labeled whole cardiomyocytes or cardiomyocyte remnants, one would not expect to detect differences in the number of labeled cells in sham-operated, infarcted, and cell-treated hearts (as well as in pre-pulsed hearts), as those conditions would not alter the level of contamination by labeled resident cardiomyocytes in our cell preparations. Finally, it should be noted that the majority of the detected labeled cells did not express sarcomeric proteins (Fig 2D), which by default excludes the possibility that they were cardiomyocyte remnants. Nevertheless, we acknowledge labeling of resident cardiomyocytes as a limitation of our study. Second, our genetic labeling approach cannot identify the identity of the parent cell (prior to activation of the αMHC promoter) that gives rise to endogenous committed cardioblasts. The investigation of surface receptor expression in cardioblasts identified Sca-1 as a potential (albeit non-specific) marker that may be expressed in parent cells. It should be noted, though, that Sca-1 (Oh et al, 2003) is expressed in several putative adult cardiac progenitors [including c-kit$^+$ (Bailey et al, 2012) and PDGFRα$^+$ (Chong et al, 2011) cells]. Ultimately, forward-fate mapping experiments for putative stem cell markers will be required to reveal the identity of the parent cell that gives rise to cardioblasts. To that end, a recent study employing forward fate mapping for Sca-1 demonstrated a contribution of Sca-1-derived cells to the myocyte pool in the normal heart, which increases post-myocardial injury (Uchida et al, 2013). Third, even though we show that endogenous cardioblasts can differentiate into mature cardiomyocytes after injection into recipient non-transgenic hearts, our fate mapping model (which also labels cardiomyocytes) does not allow for the investigation of cardiomyogenic differentiation of cardioblasts in situ or for direct quantification of the number of cardiomyocytes that arise from this cell population. Our previous study using the same fate mapping model (in which administration of 4OH-tamoxifen prior to injury was combined with long-term BrdU pulsing post-injury) provided indirect evidence that endogenous progenitors replenish approximately 1% of total myocytes post-MI and approximately 2.5% of total myocytes post-cell therapy with CDCs, during the first 5 weeks post-injury (Malliaras et al, 2013a). Fourth, even though we show that cell therapy with CDCs

upregulates endogenous cardioblasts partially through SDF1 secretion, our model cannot discern whether this is a result of increased survival, activation, proliferation, or recruitment of endogenous cardioblasts. SDF1 has been shown to affect cell homing to the injured myocardium (Abbott et al, 2004) [including recruitment of putative endogenous cardiomyocyte progenitors (Segers et al, 2007; Unzek et al, 2007)] and to exert cytoprotective (Zhang et al, 2007), pro-proliferative (Lataillade et al, 2000), and angiogenic (Segers et al, 2007; Zhang et al, 2007; Saxena et al, 2008; Kanki et al, 2011) effects (the latter were also observed in our study, Supplementary Fig S8), all of which could directly or indirectly contribute to the detection of increased numbers of endogenous cardioblasts in the infarcted area post-cell therapy. Finally, while we show a correlation between CDC-induced functional benefits and a SDF-1-dependent increase in the number of endogenous cardioblasts, our study does not unequivocally demonstrate that cardioblasts play a direct role in the benefits afforded by CDC therapy, since: (a) our fate mapping model does not allow for direct quantification of the number of cardiomyocytes that arise from GFP$^+$ cardioblasts and (b) SDF1 has pleiotropic effects (Frangogiannis, 2011; Penn et al, 2012).

## Conclusions

In summary, our study provides evidence for the activation of endogenous cardioblasts in the injured adult mammalian heart. We provide insight into the molecular identity, functional properties, and anatomic origin of said cardioblasts and identify treatment strategies and paracrine signaling pathways that could be used for therapeutic stimulation of innate progenitor cell-mediated cardiomyocyte replenishment.

## Materials and Methods

### Generation of bitransgenic MerCreMer/ZEG Mice and 4-OH-Tamoxifen pulse

All animal procedures were approved by the Cedars-Sinai Medical Center Animal Care and Use Committee (IACUC2557). Bitransgenic mice were generated by crossbreeding female transgenic B6129-Tg (Myh6-cre/Esr1)1Jmk/J (Myh6-MerCreMer) mice with male B6.Cg-Tg(ACTB-Bgeo/GFP)21Lbe/J (ZEG) reporter mice (Jackson Laboratory). Myh6-MerCreMer mice carry a fusion transgene of Cre recombinase flanked by mutated estrogen receptor ligand-binding domains, driven by the cardiac αMHC promoter (encoded by Myh6). ZEG reporter mice carry a lacZ transgene flanked by LoxP sites, followed by stop codons and then the GFP gene; therefore, upon excision of the LoxP sites and stop codons mediated by Cre recombinase activity, the reporter switches to GFP (driven by the β-actin promoter), permanently and specifically marking αMHC$^+$ cells and their progeny as GFP$^+$. Animal genotype was verified by RT–PCR on tail genomic DNA (Zhang et al, 2010; Malliaras et al, 2013a). 4-OH-tamoxifen (Sigma), dissolved in peanut oil (Sigma) at a concentration of 5 mg/ml, was intraperitoneally injected into double heterozygous bitransgenic MerCreMer/ZEG mice daily at a dose of 0.5 mg/d to induce Cre recombination.

## CDC culture

Mouse CDCs were expanded from explanted hearts obtained from 8-week-old male B6129SF1/J mice as described (Malliaras *et al*, 2013a). In brief, myocardial specimens were cut into fragments less than 1 mm$^3$ and cultured as cardiac explants on fibronectin-coated dishes. After a variable period of growth, a layer of stromal-like cells emerged from the cardiac explant over which phase bright cells proliferated. The cells surrounding the explant were harvested using enzymatic digestion with 0.05% trypsin and were seeded at 50,000 cells/ml on poly-D-lysine-coated dishes, generating cardiospheres. Three days later, free-floating cardiospheres were harvested and plated on fibronectin-coated flasks to generate CDCs.

## MI creation and cell injection

An acute MI was created in adult bitransgenic MerCreMer/ZEG and B6129SF1/J mice (8- to 12-week-old). After general anesthesia and tracheal intubation, mice were artificially ventilated with room air. A left thoracotomy was performed through the fourth intercostal space, and the left anterior descending artery was ligated with 9-0 prolene under direct vision. Mice were then subjected to intramyocardial injections with a 30-G needle at four points in the infarct border zone with 40 μl of PBS (infarcted controls), $10^4$ GFP$^+$ small cells or $2 \times 10^5$ CDCs. Mice in the sham groups underwent left thoracotomy without LAD ligation. In a subset of experiments (Fig 5B and C), $10^4$ GFP$^+$ small cells were intramyocardially injected in uninjured hearts.

## Enzymatic digestion of mouse hearts and isolation of cardiomyocyte-depleted cardiac cells

Hearts were enzymatically digested and cardiomyocyte-depleted cardiac cells were isolated as described (Pfister *et al*, 2010). Mice were anaesthetized by ketamine/dexmedetomidine. The thorax was opened and hearts were flushed by injecting 10 ml of cold PBS into the left ventricle. Hearts were excised and cleansed to remove residual blood. The infarct and peri-infarct area (in infarcted mice) or the corresponding LV area in sham-operated controls was isolated from the rest of myocardium (remote). Subsequently, both the infarct and peri-infarct area (or the corresponding LV area in sham-operated controls) and the remote myocardium were minced. Myocardial fragments were incubated with digestion solution (collagenase B 0.1%, dispase II 2.4 U/ml, 2.5 mM CaCl$_2$, 138 mM NaCl, 5.56 mM glucose, 5.4 mM KCl, 4.16 mM NaHCO$_3$) at 37°C for 30 min with frequent pipetting to ensure complete digestion. Digested cells were subsequently filtered 3 times using a 40-μm filter to ensure removal of any residual myocytes, and red blood cell lysis was performed using ACK lysis buffer. Isolated cells were filtered three additional times and resuspended in cold HBSS (supplemented with 2% FBS and 10 mM HEPES). In a subset of experiments (Fig 2A), enzymatically digested myocyte-depleted cardiac cell preparations were plated (100,000 cells/ml) in fibronectin-coated glass bottom gridded dishes (Matek) and cultured in DMEM/F12, supplemented with 20% FBS, 1% penicillin/streptomycin, and 10 ng/μl bFGF. Chemicals were purchased from Sigma-Aldrich.

## Isolation of cardiomyocytes

In a subset of experiments (Figs 3 and 5C), cardiomyocytes were isolated from bitransgenic MerCreMer (Fig 3) and B6129SF1/J (Fig 5C) mice by enzymatic dissociation of the whole heart on a Langendorff apparatus as described (Malliaras *et al*, 2013a). Heparinized animals were anaesthetized by ketamine/xylazine (30 mg/Kg and 6 mg/Kg, respectively). Hearts were rapidly excised and cleansed to remove blood in ice-cold Tyrode's solution before being mounted to a Langendorff apparatus conjugated to a pressure-monitoring device, then perfused retrogradely with the following two oxygenated solutions in sequential order: (1) Ca$^{2+}$-free Tyrode's solution (2–3 min) and (2) Ca$^{2+}$-free Tyrode's solution containing 1 mg/ml collagenase B (Roche) and 0.1 mg/ml protease (Sigma) for 10–15 min depending on digesting conditions. Hearts were subsequently dismounted, atria were discarded, and digested ventricles were cut off and minced in Kruftbrühe (KB) solution, followed by pipetting to dissociate the cells and filtered through a nylon mesh (250 μm pore size) to remove big pieces of undigested tissues. Isolated cells were rinsed in KB solution and allowed to settle by gravity to remove debris and non-cardiomyocytes. The remaining cells were then resuspended in KB solution for further experiments. Tyrode's solution contained (mM): NaCl 105, KCl 5.4, KH$_2$PO$_4$ 0.6, NaH$_2$PO$_4$ 0.6, NaHCO$_3$ 6, KHCO$_3$ 5, MgCl$_2$ 1, HEPES 10, glucose 5, taurine 20, BDM 20 (pH 7.4 with NaOH). KB solution contained (mM): KCl 20, KH$_2$PO$_4$ 10, K$^+$-glutamate 70, MgCl$_2$ 1, glucose 25, β-hydroxybutyric acid 10, taurine 20, EGTA 0.5, HEPES 10, and 0.1% albumin (pH 7.4 with KOH). Chemicals were purchased from Sigma-Aldrich.

## FACS of myocyte-depleted cardiac cells

Enzymatically digested myocyte-depleted cardiac cell preparations resuspended in HBSS (supplemented with 2% FBS and 10 mM HEPES) were sorted on a MoFlo Cell Sorter (DakoCytomation, Inc.). Cells were sorted based on cell size (forward and side scatter) to exclude cellular debris and cells with exceedingly high forward and side scatter, and GFP expression. In a subset of experiments (Fig 5), cells were also sorted based on 7-aminoactinomycin D negativity, in order to purify live GFP$^+$ myocyte-depleted cells. Non-4OH-tamoxifen-pulsed bitransgenic mice were used as controls.

## Flow cytometry

Flow cytometry was performed in myocyte-depleted cardiac cells to measure expression of GFP, Sca-1 (560595, BD Pharmingen) and percentage of side population (SP) cells. Flow cytometry for GFP expression was performed in both freshly isolated myocyte-depleted cardiac cells and fixed cells after incubation with a GFP antibody (PA1-46326, Thermo Scientific). Flow cytometric analysis of the percentage of SP cells was performed using protocols and reagents described in detail previously (Pfister *et al*, 2010). Flow cytometry experiments were performed using a benchtop flow cytometer (CyAnADP; DakoCytomation, Inc.). Gates were established by forward and side scatter to exclude cellular debris and cells with exceedingly high forward and side scatter. Non-4OH-tamoxifen-pulsed bitransgenic mice were used as controls.

Quantitative analysis was performed using Summit software (Beckman Coulter, Inc.).

### ShRNA knockdown of SDF1 and VEGF in CDCs

Knockdown of SDF1 and VEGF in mouse CDCs was performed by commercially available lentiviral vectors expressing shRNA against mouse SDF1 (sequence CCGGGATCCAAGAGTACCTGGAGAACTC-GAGTTCTCCAGGTACTCTTGGATCTTTTTTG, clone number TRCN 0000184347) and VEGF (sequence CCGGGATCAAGTTCATGGATGT CTACTCGAGTAGACATCCATGAACTTGATCTTTTTG, clone number TRCN0000066822) (MISSION lentiviral transduction particles, Sigma-Aldrich), according to the manufacturer's instructions. A non-targeting shRNA that activates RISC and the RNAi pathway, but does not target any mouse genes served as a negative control (sh-control). Mouse CDCs (plated 24 h prior to transduction) were transduced at 50–80% confluency, after the addition of hexadimethrine bromide (8 μg/ml) (MOI: 20). The next day, the medium was replaced with fresh culture medium. The next day, the medium was replaced with puromycin-containing culture medium (5 μg/ml). Puromycin-resistant CDCs were subsequently expanded (in puromycin-containing medium) until the time of the experiments. To confirm successful knockdown of SDF1 and VEGF in CDCs, shRNA-transduced CDCs were cultured in FBS-free IMDM medium for 2 days. The supernatants (conditioned media) were collected and the concentrations of SDF1 and VEGF were measured with mouse-specific ELISA kits (R&D Systems Inc.), according to the manufacturer's instructions. To investigate whether transplantation of shRNA-transduced CDCs resulted in decreased myocardial levels of SDF1 and VEGF *in vivo*, mice were sacrificed 4 days after MI and injection of shRNA-transduced CDCs. Myocardial protein was isolated from the peri-infarct area and Western blots were performed as described (Malliaras *et al*, 2012). Quantitative analysis was performed by ImageJ software, and expression levels were normalized to β-actin.

### Bone marrow transplantation

Bone marrow transplantation was performed as described (Wang *et al*, 2006). In brief, bone marrow was isolated from the femurs and tibias of donor 8-week-old MerCreMer/ZEG mice. Recipient B6129SF1/J mice (8-week-old) were lethally irradiated with 1,100 rads, given in two doses 4 h apart. Subsequently, 5 million bone marrow cells were injected into the tail vein of the recently lethally irradiated recipient mice. Two mice did not receive bone marrow transplantation and served as controls to ensure lethal irradiation (both mice died within 2 weeks post-irradiation).

### RT–qPCR

RNA was extracted from FACS-sorted $GFP^+$ and $GFP^-$ myocyte-depleted cardiac cells (20,000 cells per condition/heart, obtained from the same hearts) and from isolated cardiomyocytes using RNeasy Micro Kit (Qiagen). Two-step RT–qPCR (with 14 cycles of cDNA preamplification) was performed in 7900HT Fast real-time PCR System, using commercially available primers and probes (Taqman assays, Applied Biosystems) according to the manufacturer's instructions. The expression of the following genes was investigated: NKX2-5, MEF2C, GATA4, TBX3, FOXC2, BMP2, ISL1, TBX2, TBX5, TBX20, MESP1, MESP2, GATA6, WNT2, WNT5a, WNT11, DKK1, HAND1, HAND2, FOXC1, SMARCD3, MYH6, MYH7, MYL2, MYL3, TNNI3, TNNT2, MYBPC3, PLN, DES, NPPA, CKM, TTN, VIM, VWF, ACTA2, PDGFRa, KIT, FUT4, and CD34. GAPDH was used as the internal control. Data were analyzed using the 7900HT Sequence Detection System Software (Applied Biosystems), and fold change was calculated with the comparative CT method.

### Epifluorescence microscopy, immunocytochemistry, and histology

Enzymatically digested myocyte-depleted cardiac cell preparations resuspended in HBSS (supplemented with 2% FBS and 10 mM HEPES) were plated onto laminin-coated slides and incubated with Hoechst 33342 (H1399, Molecular probes). Thirty minutes later, the percentage of $GFP^+$ cells was quantified using epifluorescence microscopy (Nikon, ECLIPSE TE2000). Enzymatically digested myocyte-depleted cardiac cell preparations (either freshly isolated or after FACS for the purification of $GFP^+$ cells) were cytospun onto laminin-coated slides to undergo fluorescent immunocytochemistry. Fluorescent immunocytochemistry was performed with antibodies against GFP (ab13970, Abcam), α-sarcomeric actin (A7811, Sigma), α-myosin heavy chain (ab15, Abcam), NKX2-5 (ab35842, Abcam), GATA4 (ab84593, Abcam), MEF2C (ab64644, Abcam), TBX5 (ab101227, Abcam), Isl1 (ab20670, Abcam), Ki67 (RM-9106, Thermo Scientific), H3P (ab5176, Abcam), lacZ (ab9361, Abcam), Sca-1 (ab25195, Abcam), Nestin (ab6142, Abcam), c-kit (ab5506, Abcam), SSEA1 (ab16285, Abcam), CD34 (ab8158, Abcam), PDGFRα (ab61219, Abcam), CD31 (ab28364, Abcam), CD 45 (ab10558, Abcam), CXCR4 (ab2074, Abcam), VEGFR1 (ab32152, Abcam), VEGFR2 (ab2349, Abcam), and VEGFR3 (ab27278, Abcam). For histology, hearts were arrested with KCl solution, explanted, frozen in OCT compound, and sectioned in 6-μm sections on a cryostat. Cryosections were subsequently fixed with 4% para-formaldehyde. Quantitative morphometric analysis with Masson's trichrome staining was performed on 6-μm sections (four sections per heart at 500-μm intervals, starting at the level of LAD ligation). For each section, infarct size was determined as the percentage of LV area by manual tracing (ImageJ). Scar and viable myocardium volumes were calculated by multiplying scar and viable area for each slice with 500 μm (the sectioning interval). Scar and viable mass were quantified by multiplying total LV mass (measured at the time of heart explantation) with the volumetric fraction of scar and viable myocardium (expressed as percentage of total LV volume). Sections were stained with α-sarcomeric actinin (A7811, Sigma), GFP (ab13970, Abcam), lacZ (ab9361, Abcam), isolectin B4 (I21411, Molecular probes), connexin 43 (C6219, Sigma), CD45 (ab10558, Abcam), von Willebrand factor (ab11713, Abcam), α-smooth muscle actin (ab5694, Abcam), Ki67 (RM-9106, Thermo Scientific), NKX2-5 (ab35842, Abcam), GATA4 (ab84593, Abcam), and MEF2C (ab64644, Abcam). Border zone (Fig 8G) was defined as the area within one 20×-power field from the edges of the scar. In all fluorescent immunocytochemistry and immunohistochemistry experiments, Alexa Fluor-conjugated secondary antibodies (Molecular probes) were used and counterstaining with DAPI (P36931, Molecular probes) was performed. Sections were imaged using a confocal laser scan microscope (Leica Microsystems), and images were processed by Leica LAS software suite.

### The paper explained

#### Problem
The identity, properties, and physiologic role of endogenous cardio-myocyte progenitors in the adult mammalian heart are unknown.

#### Results
We demonstrate that the adult mouse heart contains endogenous cardiomyocyte progenitors (cardioblasts); these cells express cardiac transcription factors and sarcomeric proteins, exhibit spontaneous contractions, and form mature cardiomyocytes *in vivo*. Endogenous cardioblasts are rarely evident in the normal adult mouse heart. However, myocardial infarction results in significant cardioblast activation in the site of injury. The activated progenitors do not arise from hematogenous seeding, cardiomyocyte dedifferentiation, or mere expansion of a preformed progenitor pool. Cell therapy with cardiosphere-derived cells amplifies innate cardioblast-mediated tissue regeneration, in part through the secretion of stromal cell-derived factor 1.

#### Impact
We provide insight into endogenous cardiomyocyte progenitors in the adult mammalian heart and identify treatment strategies and para-crine signaling pathways that could be used for therapeutic stimulation of innate progenitor cell-mediated cardiac regeneration.

## Echocardiography

Mice underwent echocardiography at 24 h (baseline) and 5 weeks after surgery using Vevo 770TM Imaging System (VISUALSONICS) as described (Malliaras *et al*, 2013a). LVEF, LV end-diastolic volume, and LV end-systolic volume were measured with Visual-Sonics V1.3.8 software from 2D long-axis views taken through the infarcted area at the level of the greatest LV diameter.

## Statistics

Results both in text and in figures are presented as means $\pm$ SD. Statistical significance was determined by independent samples *t*-test, non-parametric Mann–Whitney *U*-test, and one-way ANOVA followed by LSD *post hoc* test (SPSS 16.0). Differences were considered significant when $P < 0.05$. The exact *P*-values and the specific statistical test performed for each experiment are provided in Supplementary Table S1. All experiments were open-label. All measurements were performed in a non-blinded fashion. The number of replicates used in each experiment is indicated in the figure legends.

**Supplementary information** for this article is available online: http://embomolmed.embopress.org

## Acknowledgements
We thank Patricia Lin for assistance with FACS and flow cytometry and Dorothy Khan for breeding the bitransgenic mice. This work was supported by NIH (R01 HL083109), the California Institute for Regenerative Medicine, and the Cedars-Sinai Board of Governors Heart Stem Cell Center.

## Author Contributions
KM established the hypotheses, designed the study, performed experiments, analyzed data, and wrote the paper. AI assisted with immunocytochemis-try, immunohistochemistry, and histology. ET assisted with 4OH-tamoxifen pulsing of mice. WL performed immunocytochemistry, immunohistochemistry, and animal genotyping. BS performed mouse surgeries. RCM performed RT-qPCR, immunocytochemistry, immunohistochemistry, histology, ELISA and analyzed data. JS performed flow cytometry, epifluorescence microscopy, immunocytochemistry, immunohistochemistry, histology, animal genotyping, RNA isolation and analyzed data. LW designed the bone marrow transplantation experiments and performed irradiation of mice and bone marrow transplantation. BGS designed the bone marrow transplantation experiments. EM established the hypotheses, designed the study, co-analyzed data, and wrote the paper.

## Conflict of interest
EM is a founder and equity holder in Capricor, Inc. KM has received consulting fees from Capricor, Inc. Capricor, Inc., provided no funding for this study. The remaining authors declare that they have no conflict of interest.

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
