## [Review Process File · EMBO Molecular Medicine]

Stimulation of endogenous cardioblasts by exogenous cell therapy after myocardial infarction

Konstantinos Malliaras, Ahmed Ibrahim, Eleni Tseliou, Weixin Liu, Baiming Sun, Ryan Middleton, Jeffrey SeKonstantinos Malliaras, Ahmed Ibrahim, Eleni Tseliou, Weixin Liu, Baiming Sun, Ryan C. Middleton, Jeffrey Seinfeld, Lai Wang, Behrooz G. Sharifi, Eduardo Marb·ninfeld, Lai Wang, Behrooz Sharifi

Corresponding author: Eduardo Marban, Cedars-Sinai Heart Institute

Review timeline:	Submission date:	30 October 2013
	Editorial Decision:	12 December 2013
	Revision received:	18 March 2014
	Accepted:	27 March 2014

Transaction Report:

Editor: Céline Carret

1st Editorial Decision

12 December 2013

Thank you for the submission of your manuscript to EMBO Molecular Medicine. We have now heard back from the three referees whom we asked to evaluate your manuscript. Although the referees find the study to be of potential interest, they also raise a number of concerns that must be addressed in a major revision of your work.

As you will see from the referees' comments below, while referees 1 and 2 are more supportive of the study, referee 3 is rather negative. We would like to ask you to focus on convincingly addressing all concerns of referees 1 and 2 experimentally when required. We also would strongly encourage you to reply to the specific critique of reviewer #3 as it partly overlaps with the raised concerns of the other reviewers ("Why do the GFP+ CPC-derived cells not survive in culture? How can the authors rule out that the rare beating cell reported is not simply carryover of a small mono-nucleated cardiomyocyte? How can the authors rule out that the very rare highly differentiated cardiomyocytes following engraftment of CPCs did not arise from fusion events?").

Please note that it is EMBO Molecular Medicine policy to allow a single round of revision in order to avoid the delayed publication of research findings. Consequently, acceptance or rejection of the manuscript will depend on the completeness of your responses included in the next version of the manuscript.

EMBO Molecular Medicine has a "scooping protection" policy, whereby similar findings that are published by others during review or revision are not a criterion for rejection. Should you decide to submit a revised version, I do ask that you get in touch after three months if you have not completed

it, to update us on the status.

I look forward to receiving your revised manuscript.

***** Reviewer's comments *****

Referee #1 (Comments on Novelty/Model System):

I ranked the novelty of this study to be "medium" because this study is just another addition to the previous study of theirs and others that utilized *ex vivo* expanded progenitors cells being injected into the host mice.

Referee #1 (Remarks):

In this manuscript, Malliaras et al. demonstrated the existence of cardiac progenitor cells using the previously characterized transgenic mice but employing a different strategy than the previous study of the authors. The authors isolated these progenitor cells and characterized them *in vitro* and *in vivo*. Also, they performed functional studies by targeting SDF1 to comment on a possible therapeutic approach using these identified cells.

This study has been carefully conducted using various methods to overcome the limitation of the mouse model that the authors employed (e.g. labeling of pre-existing cardiomyocytes), which are described in detail in the main text.

Overall, this study confirms the existence of cardiac progenitor cells in the murine heart, which can be activated upon injury to participate for the limited regeneration of the heart.

My comments are as follows:

(1) "GFP+ CPCs were highly enriched in the infarct and peri-infarct area (Fig 1F), compared to the remote myocardium (Fig 1G,H)."

From the images, it is not evident where is the area of the infarct. It is desirable that the authors indicate such area by performing a staining such as using an anti-Collagen antibody.

(2) All the confocal images are magnified to a certain extent. As the authors claim in text that the upon MI, the number of GFP+ CPC increases compared to sham-operated hearts. Although the authors demonstrate such increase using FACS, it is desirable to see such increased evidence on the heart sections using immunohistochemistry. In other words, lower magnification images should be provided as supplementary figures.

(3) "While few GFP+ CPCs survived in culture, ~11% (7/66) exhibited weak spontaneous contractions *in vitro* over the course of the first 2 days post-plating, without exposure to cardiac differentiation medium (Fig 2A, Supp Videos 1,2). Modifications in the culture conditions (including plating isolated cardiac cell preparations onto a feeder layer, or culturing cells in embryonic stem cell medium) failed to improve survival of GFP+ CPCs *in vitro*. While rare examples of dividing GFP+ CPCs could be detected (Supp Fig 5), no GFP+ CPCs survived more than 1 week in culture."

Although the authors provided the results of staining as well as movies, spontaneous contraction after 2 days in culture seems very quick. Also, these cells die off within one week in the culture. Previous studies by a number of laboratories demonstrated much longer time in the culture for their isolated CPCs to differentiate into beating cardiomyocytes *ie.g.* <http://www.jbc.org/content/279/12/11384.long>. This point needs to be discussed extensively.

(4) "To investigate expression of cardiogenic molecules and cardiac structural factors at the transcript level, we performed RT-qPCR in RNA extracted from FACS-sorted GFP+ and

GFPmyocyte-depleted cardiac cells (obtained from the same hearts, n=5 hearts) and from adult cardiomyocytes (n=3) for genes that are upregulated during cardiomyogenic differentiation of embryonic stem cells (Paige et al, 2012)."

How many cells were used to prepare total RNAs for each condition? Based on the results of Figure 2, it seems that the CPCs that the authors isolated are rather heterogeneous population of cells.

(5) "Immunocytochemistry demonstrated that GFP+ CPCs were negative for c-kit, SSEA1, PDGFR and CD34 (Fig 4D); however, we cannot exclude the possibility that enzymatic tissue digestion altered the expression of surface markers via cleavage or that stem-cell related antigens may be lost as progenitors begin to differentiate."

If the authors would like to argue this way, then they should perform RT-PCR for c-kit, SSEA1, PDGFR and CD34 to see whether enzymatic digestion altered the expression of the above mentioned surface markers.

(6) "We then injected these CPCs into normal (n=3) or infarcted (n=3) recipient hearts of background non-transgenic mice (10,000 CPCs/heart, Fig 5A)...Rare events of cardiomyogenic differentiation could be detected in four of six CPC-injected hearts (2/3 normal and 2/3 infarcted hearts; Fig 5B-D)."

Any quantifiable numbers for such rare events?

(7) It is written as follows in Pages 10-11:

"To investigate whether the increase in GFP+ CPCs observed post-MI originates from dedifferentiation of resident myocytes or from expansion of a pre-existing pool of already committed (MHC+) progenitors post-injury, non-infarcted bitransgenic mice underwent daily 4OH-tamoxifen pulsing for 10 days...Thus, the majority of GFP+ CPCs are activated (i.e. turn on the MHC promoter) post-MI, although a small minority may originate from expansion of a preexisting, already-committed CPC pool or from dedifferentiation of resident myocytes."

However, in the Discussion section, it is written as follows:

"With regard to their anatomical origin, we demonstrate that the identified endogenous CPCs do not arise from hematogenous seeding, dedifferentiation of mature cardiomyocytes or mere expansion of a preformed pool of already-committed MHC+ CPCs."

I think "do not arise" is rather a strong statement unless the authors provide a direct evidence for this. In addition, the authors should discuss the reason behind not using BrdU pulse-and-chase experiments after the MI as the authors have done in their previous publication, which they performed tamoxifen and BrdU injections prior to the MI injury.

(8) "Ultimately, forward-fate mapping experiments for putative stem cell markers (including Sca-1) will be required to reveal the identity of the parent cell that gives rise to CPCs."

A recent study by Uchida et al. ([http://www.cell.com/stem-cell-reports/abstract/S2213-6711\(13\)00091-X](http://www.cell.com/stem-cell-reports/abstract/S2213-6711(13)00091-X)) reported the above mentioned method.

(9) The catalog number of each antibody employed in this study should be provided.

Referee #2 (Remarks):

The study uses the tamoxifen-dependent MHCalpha-MerCreMer x ZEG genetic lineage tracing system to detect cardiac progenitor cells in the murine heart. In contrast to previous studies using this system, which relied on a dilution of labeled myocytes with aging or infarction to make the case for a contribution of CPCs, these authors focus on small mononuclear non-myocyte cells that express MHCalpha-GFP as a readout of CPC presence and expansion. Thus, labeling is performed (in some experiments) after MI rather than before MI as in previous studies. This approach detects immature cardioblasts, but not uncommitted cardiac progenitor cells. One key issue in the work is the possibility for confusing cardiomyocyte fragments with CPCs. Some effort has gone into minimizing cardiomyocyte fragment contamination, although this is acknowledged as a limitation.

The study finds that CPCs as defined above are rare in the heart (0.12% of non-myocyte cells) but increase ~10x after MI being enriched in the infarct and peri-infarct areas. The cells are GFP+ and Sca1+ (but ckit-, CD45- and Pdgfra-) and of cardiac TFs express Nkx2-5 and Gata4 but not Tbx5 or Isl1. They express high levels of myosins, troponins etc but not markers of SM or ECs. Few survive in cultured but a fraction of those that do can undergo spontaneous contractions. After injection into normal or infarcted recipients, rare cells could be identified as cardiomyocytes by immunohistochemistry. They did not arise apparently from bone marrow as shown by labeled BM transplantation after lethal irradiation.

The argument was made, based on experiments in which tamoxifen was pulsed pre- and post-MI, that most of these CPCs are created from cells that do not express the transgene, and therefore are likely to be true cardiac progenitor cells (and not de-differentiated myocytes).

The authors go on to study how cell therapy at the time of MI using cardiosphere-derived cells might influence this CPC population. They show upregulation in a model in which sham operated mice were the control (with no control cells being injected). The therapy decreased scar mass and increased viable myocardium, as well as increased capillary density and ejection fraction. Small hairpin RNA knockdown of SDF-1 partially reversed these effects.

Strengths.

1. This is a careful study that adds further weight to the argument for the presence of undifferentiated stem/progenitor cells that participate in cardiac repair after MI. The alternative use of the lineage tracing tools is a strength since it directly measures immature cardioblasts, albeit that their origins from undifferentiated CPCs is inferred.
2. The study also adds strength to the realization that most cell therapies currently being trialed, including those with CDCs, have their beneficial effect by conditioning the endogenous repair processes, including the expansion of CPCs.

Weaknesses.

1. The possibility that cardioblasts are being confused with cardiomyocyte fragments remains, although has been clearly addressed at the technical level and as a caveat of the study.
2. The study demonstrates a correlation between benefits afforded by CDC therapy and increased number of CPCs, which are SDF-1 dependent, although falls short of demonstrating that CPCs play any significant role in those benefits. This should be very clear in the Discussion.
3. I doubt that the cells should be called CPCs since they express MHC and a variety of other sarcomeric genes and do not show differentiation in vivo to vascular or SM structure. At best they should be called putative cardioblasts with origins from a CPC inferred.

Other comments:

1. Page 5. Sentence at end of para 1.: "These result demonstrate the MI results in CPC activation and recruitment to risk area". Recruitment cannot be claimed based on current data.
2. Fig. 2F: This type of analysis cannot be used to quantify cell division without costaining with aurora B kinase attachment to the mitotic spindle.
3. The origin of these cells is not clear as they do not express ckit (CPC) or Pdgfra (CFU-F/MSC) - Sca1 is expressed on both of these populations. This should be made clearer in Discussion.
4. Page 9. The use of an historical control group is inappropriate, even when acknowledging that a functional experiment was not performed. This should be deleted.

Referee #3 (Remarks):

The manuscript by Malliaras et al uses a previously described conditional activation model (comprising a tamoxifen-dependent cre driver in combination with the ZEG reporter) to attempt to demonstrate the presence of endogenous cardiomyocyte progenitors in mouse hearts. The authors perform a questionable technical variation with the model (namely, induction of tamoxifen-mediated

reporter recombination after myocardial injury rather than before) in the hopes of identifying progenitors which are in the process of activating the cardiomyogenic progenitor program. The study is fatally flawed at several levels. First, the authors state that induction of recombination in this model prior to myocardial injury has demonstrated the presence of cardiomyogenic stem cells as evidenced by dilution of the green cardiomyocyte content over time, and further go on to perform such experiments following CDC transplantation. While the Lee laboratory has reported that this model demonstrates stem cell-derived cardiomyocyte renewal (in the Nature Genetics and Cell Stem Cell articles cited by the authors), the most recent data from the same group (Lee) published in Nature earlier this year - using the same genetic model but now in addition with stable isotope incorporation - rather convincingly indicated that cardiomyocyte renewal in the normal and injured adult mouse heart occurs via proliferation of pre-existing cardiomyocytes. Given this, the entire premise for the use of this model in the current study is greatly flawed. Second, the author's interpretation of their studies wherein reporter gene recombination is initiated after myocardial injury is dependent upon ABSOLUTE elimination of differentiated cardiomyocytes in the digestion/isolation steps. This is not likely using the protocols used, and certainly was not documented satisfactorily. The authors have also recently shown (in EMBO Medicine) that CDC delivery promotes cardiac myocyte renewal via two mechanisms - 50% via direct differentiation of the CDCs into de novo cardiomyocytes, and 50% via proliferation of pre-existing cardiomyocytes. Given the overtly quantitative nature of that earlier study, there is no additional wiggle room to account for the additional mechanisms for CDC-mediated renewal as was proposed here. There are numerous other concerns with the work. Why do the GFP+ CPC-derived cells not survive in culture? How can the authors rule out that the rare beating cell reported is not simply carryover of a small mono-nucleated cardiomyocyte? How can the authors rule out that the very rare highly differentiated cardiomyocytes following engraftment of CPCs did not arise from fusion events? The data using BMT to demonstrate that the marrow does not carry cells with cardiomyogenic activity has been previously shown using the same model by Lee's group.

1st Revision - authors' response

18 March 2014

Referee #1 (Comments on Novelty/Model System):

I ranked the novelty of this study to be "medium" because this study is just another addition to the previous study of theirs and others that utilized ex vivo expanded progenitors cells being injected into the host mice.

Referee #1 (Remarks):

In this manuscript, Malliaras et al. demonstrated the existence of cardiac progenitor cells using the previously characterized transgenic mice but employing a different strategy than the previous study of the authors. The authors isolated these progenitor cells and characterized them in vitro and in vivo. Also, they performed functional studies by targeting SDF1 to comment on a possible therapeutic approach using these identified cells.

This study has been carefully conducted using various methods to overcome the limitation of the mouse model that the authors employed (e.g. labeling of pre-existing cardiomyocytes), which are described in detail in the main text.

Overall, this study confirms the existence of cardiac progenitor cells in the murine heart, which can be activated upon injury to participate for the limited regeneration of the heart.

My comments are as follows:

(1) "GFP+ CPCs were highly enriched in the infarct and peri-infarct area (Fig 1F), compared to the remote myocardium (Fig 1G,H). "

From the images, it is not evident where is the area of the infarct. It is desirable that the authors indicate such area by performing a staining such as using an anti-Collagen antibody.

The area of the infarct in the 2 micrographs in Fig 1F is the area that does not contain cardiomyocytes (negative for α SA) but contains cells (visualized by DAPI nuclear staining), i.e. the area that appears “black” (non-red) and contains cells (blue nuclei). This is now explained in the figure legend.

In the revised manuscript (page 44, line 18) we now state: “*The infarcted area is identified by the lack of cardiomyocytes (negative for α SA) and the presence of non-myocyte (α SA-/DAPI+) cells.*”

(2) *All the confocal images are magnified to a certain extent. As the authors claim in text that upon MI, the number of GFP+ CPC increases compared to sham-operated hearts. Although the authors demonstrate such increase using FACS, it is desirable to see such increased evidence on the heart sections using immunohistochemistry. In other words, lower magnification images should be provided as supplementary figures.*

We chose to include high magnification images in our manuscript to allow for better visualization of GFP+ cardioblasts. The identified cardioblasts are small (~11.5 μ m in diameter) and their GFP fluorescence is rather difficult to visualize in low magnification images. Nevertheless, based on the reviewer comments, in the supplement of the revised manuscript we now provide images that cover a much larger field, showing several GFP+ cardioblasts in the infarct area (Supp Fig 3).

(3) *"While few GFP+ CPCs survived in culture, ~11% (7/66) exhibited weak spontaneous contractions in vitro over the course of the first 2 days post-plating, without exposure to cardiac differentiation medium (Fig 2A, Supp Videos 1,2). Modifications in the culture conditions (including plating isolated cardiac cell preparations onto a feeder layer, or culturing cells in embryonic stem cell medium) failed to improve survival of GFP+ CPCs in vitro. While rare examples of dividing GFP+ CPCs could be detected (Supp Fig 5), no GFP+ CPCs survived more than 1 week in culture."*

Although the authors provided the results of staining as well as movies, spontaneous contraction after 2 days in culture seems very quick. Also, these cells die off within one week in the culture. Previous studies by a number of laboratories demonstrated much longer time in the culture for their isolated CPCs to differentiate into beating cardiomyocytes I.e.g.

<http://www.jbc.org/content/279/12/11384.long>. This point needs to be discussed extensively.

We thank the reviewer for this comment. The fact that the identified GFP+ cardioblasts (in contrast to other putative stem cell populations reported previously) exhibited spontaneous contractions: a) early in culture, and b) without exposure to cardiomyogenic medium, can likely be attributed to the fact that they represent already-committed, partially-differentiated myocyte progenitors, rather than uncommitted, undifferentiated parent stem cells. This is now discussed in the revised manuscript.

On page 15, line 23, we now state: “*The identified GFP+ cardioblasts exhibited spontaneous contractions early in culture without exposure to cardiomyogenic medium, in contrast to previously-reported putative stem cell populations (Matsuura et al, 2004; Jesty et al, 2012); this difference may reflect the fact that our cells represent already-committed, partially-differentiated myocyte progenitors, rather than uncommitted, undifferentiated parent stem cells.*”

(4) *"To investigate expression of cardiogenic molecules and cardiac structural factors at the transcript level, we performed RT-qPCR in RNA extracted from FACS-sorted GFP+ and GFP-myocyte-depleted cardiac cells (obtained from the same hearts, n=5 hearts) and from adult cardiomyocytes (n=3) for genes that are upregulated during cardiomyogenic differentiation of embryonic stem cells (Paige et al, 2012)."*

How many cells were used to prepare total RNAs for each condition? Based on the results of Figure 2, it seems that the CPCs that the authors isolated are rather heterogeneous population of cells.

RNA was isolated from 20,000 cells for each sample. This is now clarified in the revised methods section. In the revised manuscript (page 26, line 16) we now state: “*RNA was extracted from FACS-*

sorted GFP+ and GFP- myocyte-depleted cardiac cells (20,000 cells per condition/heart, obtained from the same hearts)..."

(5) "Immunocytochemistry demonstrated that GFP+ CPCs were negative for c-kit, SSEA1, PDGFR α , and CD34 (Fig 4D); however, we cannot exclude the possibility that enzymatic tissue digestion altered the expression of surface markers via cleavage or that stem-cell related antigens may be lost as progenitors begin to differentiate."

If the authors would like to argue this way, then they should perform RT-PCR for c-kit, SSEA1, PDGFR α and CD34 to see whether enzymatic digestion altered the expression of the above mentioned surface markers.

We thank the reviewer for this comment. Based on the reviewer's suggestion we have performed RT-PCR for c-kit, SSEA, PDGFR α and CD34 in RNA extracted from GFP+ cardioblasts obtained from 5 hearts. The results are presented in Supp Fig 7. Expression of c-kit and SSEA1 was not detected in any samples. Expression of PDGFR α and CD34 was detected in 3/5 samples, albeit at very low levels (on average >1000 fold less compared to expression of α MHC in the same samples).

In the revised manuscript (page 8, line 13), we now state: "*Immunocytochemistry demonstrated that GFP+ cardioblasts were negative for c-kit, SSEA1, PDGFR α and CD34 (Fig 4D). To exclude the possibility that enzymatic tissue digestion altered the expression of surface markers via cleavage of stem-cell related antigens, we performed RT-qPCR in RNA extracted from FACS-sorted GFP+ myocyte-depleted cardiac cells (obtained from 5 hearts) for expression of KIT, FUT4 (SSEA1), PDGFR α and CD34. KIT and FUT4 (SSEA1) were not detected in any samples. Very low expression of PDGFR α and CD34 was detected in 3/5 samples (on average >1000 fold less compared to expression of MYH6 [α MHC] in the same samples) (Supp Fig 7). However, we cannot exclude the possibility that expression of stem-cell related antigens may be lost as progenitors begin to differentiate.*"

(6) "We then injected these CPCs into normal (n=3) or infarcted (n=3) recipient hearts of background non-transgenic mice (10,000 CPCs/heart, Fig 5A)...Rare events of cardiomyogenic differentiation could be detected in four of six CPC-injected hearts (2/3 normal and 2/3 infarcted hearts; Fig 5B-D)."

Any quantifiable numbers for such rare events?

Based on the reviewer's suggestion we now provide numbers for the events of cardiomyogenic differentiation (as detected by immunohistochemistry). In the revised manuscript (page 9, line 8) we now state: "*Rare events of cardiomyogenic differentiation could be detected in four of six cardioblast-injected hearts (2/3 normal [4.2 \pm 2.2 GFP+ cardiomyocytes/ heart section] and 2/3 infarcted [3.4 \pm 3.6 GFP+ cardiomyocytes/ heart section] hearts; Fig 5B-D).*"

(7) It is written as follows in Pages 10-11: "To investigate whether the increase in GFP+ CPCs observed post-MI originates from dedifferentiation of resident myocytes or from expansion of a pre-existing pool of already-committed (α MHC+) progenitors post-injury, non-infarcted bitransgenic mice underwent daily 4OH-tamoxifen pulsing for 10 days...Thus, the majority of GFP+ CPCs are activated (i.e. turn on the α MHC promoter) post-MI, although a small minority may originate from expansion of a preexisting, already-committed CPC pool or from dedifferentiation of resident myocytes."

However, in the Discussion section, it is written as follows: "With regard to their anatomical origin, we demonstrate that the identified endogenous CPCs do not arise from hematogenous seeding, dedifferentiation of mature cardiomyocytes or mere expansion of a preformed pool of already-committed α MHC+ CPCs."

I think "do not arise" is rather a strong statement unless the authors provide a direct evidence for this.

We thank the reviewer for this comment. While we provide evidence that endogenous cardioblasts do not arise from hematogenous seeding (based on our bone marrow transplant experiments), we do not provide direct evidence that they do not arise from dedifferentiation of mature cardiomyocytes or mere expansion of a preformed pool of already-committed α MHC⁺ cardioblasts.

Based on the reviewer's suggestion, in the revised manuscript we have modified the sentence in question, as follows (page 16, line 4): "*With regard to their anatomical origin, we demonstrate that the identified endogenous cardioblasts do not arise from hematogenous seeding. While we show that the majority of cardioblasts are activated (i.e. turn on the α MHC promoter) post-MI, a small minority may originate from pre-existing, already-committed cardioblast pool or from dedifferentiation of resident myocytes.*"

In addition, the authors should discuss the reason behind not using BrdU pulse-and-chase experiments after the MI as the authors have done in their previous publication, in which they performed tamoxifen and BrdU injections prior to the MI injury.

The goal of our previous publication was to determine the contributions of cardiomyocyte proliferation and cardiomyogenic differentiation of endogenous stem cells in postnatal cardiomyogenesis. In order to achieve this, and since we used bitransgenic mice pulsed with 4OH-Tamoxifen prior to injury, it was *imperative* to implement a BrdU pulsing approach; the only way to (indirectly and retrospectively) calculate the absolute rates and relative magnitudes of cardiomyocyte proliferation and differentiation of endogenous stem cells was by comparing the rates of BrdU incorporation in GFP⁺ versus GFP⁻ cardiomyocytes.

In the present manuscript, we instead sought to achieve prospective genetic labeling of committed endogenous progenitors in order to directly investigate the existence, identity and origin of cardiomyocyte progenitors in the injured adult mouse heart, and how these progenitors are influenced by exogenous cell therapy. Our approach (4OH-Tamoxifen pulsing post-injury combined with myocyte depletion cell isolation protocols) enabled direct identification of committed progenitors/cardioblasts (which could be readily differentiated from endogenous myocytes/myocyte fragments based on size, morphology and properties). While BrdU pulsing would provide more information with regard to active cell cycling of endogenous CPCs (which in the present manuscript was investigated by staining for Ki67 and H3P), it was unnecessary/redundant for accurate identification/labeling of cardioblasts and thus was not employed in the present study.

(8) "Ultimately, forward-fate mapping experiments for putative stem cell markers (including Sca-1) will be required to reveal the identity of the parent cell that gives rise to CPCs." A recent study by Uchida et al. ([http://www.cell.com/stem-cell-reports/abstract/S2213-6711\(13\)00091-X](http://www.cell.com/stem-cell-reports/abstract/S2213-6711(13)00091-X)) reported the above mentioned method.

We thank the reviewer for this comment. Based on the reviewer's suggestion, in the revised manuscript (page 18, line 13) we now state: "*Ultimately, forward-fate mapping experiments for putative stem cell markers will be required to reveal the identity of the parent cell that gives rise to cardioblasts. To that end, a recent study employing forward fate mapping for Sca-1 demonstrated a contribution of Sca-1-derived cells to the myocyte pool in the normal heart, which increases post-myocardial injury (Uchida et al, 2013).*"

(9) The catalog number of each antibody employed in this study should be provided.

Based on the reviewer's suggestion, the catalog number of each antibody is now stated in the methods section of the revised manuscript.

Referee #2 (Remarks):

The study uses the tamoxifen-dependent MHCalpha-MerCreMer x ZEG genetic lineage tracing system to detect cardiac progenitor cells in the murine heart. In contrast to previous studies using this system, which relied on a dilution of labeled myocytes with aging or infarction to make the case for a contribution of CPCs, these authors focus on small mononuclear non-myocyte cells that express MHCalpha-GFP as a readout of CPC presence and expansion. Thus, labeling is performed (in some experiments) after MI rather than before MI as in previous studies. This approach detects immature cardioblasts, but not uncommitted cardiac progenitor cells. One key issue in the work is the possibility for confusing cardiomyocyte fragments with CPCs. Some effort has gone into minimizing cardiomyocyte fragment contamination, although this is acknowledged as a limitation.

The study finds that CPCs as defined above are rare in the heart (0.12% of non-myocyte cells) but increase ~10x after MI being enriched in the infarct and peri-infarct areas. The cells are GFP+ and Sca1+ (but ckit-, CD45- and Pdgfra-) and of cardiac TFs express Nkx2-5 and Gata4 but not Tbx5 or Isl1. They express high levels of myosins, troponins etc but not markers of SM or ECs. Few survive in cultured but a fraction of those that do can undergo spontaneous contractions. After injection into normal or infarcted recipients, rare cells could be identified as cardiomyocytes by immunohistochemistry. They did not arise apparently from bone marrow as shown by labeled BM transplantation after lethal irradiation.

The argument was made, based on experiments in which tamoxifen was pulsed pre- and post-MI, that most of these CPCs are created from cells that do not express the transgene, and therefore are likely to be true cardiac progenitor cells (and not de-differentiated myocytes).

The authors go on to study how cell therapy at the time of MI using cardiosphere-derived cells might influence this CPC population. They show upregulation in a model in which sham operated mice were the control (with no control cells being injected). The therapy decreased scar mass and increased viable myocardium, as well as increased capillary density and ejection fraction. Small hairpin RNA knockdown of SDF-1 partially reversed these effects.

Strengths.

1. This is a careful study that adds further weight to the argument for the presence of undifferentiated stem/progenitor cells that participate in cardiac repair after MI. The alternative use of the lineage tracing tools is a strength since it directly measures immature cardioblasts, albeit that their origins from undifferentiated CPCs is inferred.

2. The study also adds strength to the realization that most cell therapies currently being trialed, including those with CDCs, have their beneficial effect by conditioning the endogenous repair processes, including the expansion of CPCs.

Weaknesses.

1. The possibility that cardioblasts are being confused with cardiomyocyte fragments remains, although has been clearly addressed at the technical level and as a caveat of the study.

We agree with the reviewer. This limitation is extensively discussed in the manuscript (pages 17-18).

2. The study demonstrates a correlation between benefits afforded by CDC therapy and increased number of CPCs, which are SDF-1 dependent, although falls short of demonstrating that CPCs play any significant role in those benefits. This should be very clear in the Discussion.

We thank the reviewer for this comment. This limitation is now clearly acknowledged in the discussion of the revised manuscript.

On page 19, line 12 we now state: *“Finally, while we show a correlation between CDC-induced functional benefits and a SDF-1-dependent increase in the number of endogenous cardioblasts, our study does not unequivocally demonstrate that cardioblasts play a direct role in the benefits afforded by CDC therapy, since: a) our fate mapping model does not allow for direct quantification of the number of cardiomyocytes that arise from GFP+ cardioblasts, and b) SDF1 has pleiotropic effects (Frangogiannis, 2011; Penn et al, 2012).”*

3. I doubt that the cells should be called CPCs since they express MHC and a variety of other sarcomeric genes and do not show differentiation in vivo to vascular or SM structure. At best they should be called putative cardioblasts with origins from a CPC inferred.

Based on the reviewer’s suggestion we now refer to the identified α MHC+ non-myocyte cells as cardioblasts throughout the manuscript (including the title).

Other comments:

1. Page 5. Sentence at end of para1.: "These result demonstrate the MI results in CPC activation and recruitment to risk area". Recruitment cannot be claimed based on current data.

Based on the reviewer’s suggestion we refrain from claiming that activated cardioblasts are recruited to the risk area throughout the manuscript (including the title).

2. Fig. 2F: This type of analysis cannot be used to quantify cell division without costaining with aurora B kinase attachment to the mitotic spindle.

Based on the reviewer’s suggestion, we no longer refer to GFP+ cardioblasts in Fig 2F and in Supp Fig 5B as “mitotic”. We instead refer to them as “GFP+ cardioblasts appearing to undergo mitosis”.

3. The origin of these cells is not clear as they do not express ckit (CPC) or Pdgfra (CFU-F/MSC) - Sca1 is expressed on both of these populations. This should be made clearer in Discussion.

We thank the reviewer for this comment. In the revised limitations paragraph of the discussion (page 18, line 8) we now state: *“Second, our genetic labeling approach cannot identify the identity of the parent cell (prior to activation of the α MHC promoter) that gives rise to endogenous committed cardioblasts. The investigation of surface receptor expression in cardioblasts identified Sca-1 as a potential (albeit non-specific) marker that may be expressed in parent cells. It should be noted, though, that Sca-1 (Oh et al, 2003) is expressed in several putative adult cardiac progenitors (including c-kit+ [Bailey et al, 2009] and PDGFR α + [Chong et al, 2011] cells).”*

4. Page 9. The use of an historical control group is inappropriate, even when acknowledging that a functional experiment was not performed. This should be deleted.

Based on the reviewer’s suggestion, we have deleted the reference to an historical control group. In the revised manuscript (page 9, line 18) we now state: *“Even though we did not perform a controlled study to investigate whether transplantation of GFP+ cardioblasts confers therapeutic benefit to infarcted mice, comparison of cardiac function by echocardiography between baseline and 5 weeks post-MI revealed a significant decline in ejection fraction (30 \pm 5% vs 22 \pm 7%, p=0.036).”*

Referee #3 (Remarks):

The manuscript by Malliaras et al uses a previously described conditional activation model (comprising a tamoxifen-dependent cre driver in combination with the ZEG reporter) to attempt to demonstrate the presence of endogenous cardiomyocyte progenitors in mouse hearts. The authors perform a questionable technical variation with the model (namely, induction of tamoxifen-mediated reporter recombination after myocardial injury rather than before) in the hopes of identifying progenitors which are in the process of activating the cardiomyogenic progenitor program. The study is fatally flawed at several levels.

First, the authors state that induction of recombination in this model prior to myocardial injury has demonstrated the presence of cardiomyogenic stem cells as evidenced by dilution of the green cardiomyocyte content over time, and further go on to perform such experiments following CDC transplantation. While the Lee laboratory has reported that this model demonstrates stem cell-derived cardiomyocyte renewal (in the Nature Genetics and Cell Stem Cell articles cited by the authors), the most recent data from the same group (Lee) published in Nature earlier this year - using the same genetic model but now in addition with stable isotope incorporation - rather convincingly indicated that cardiomyocyte renewal in the normal and injured adult mouse heart occurs via proliferation of pre-existing cardiomyocytes. Given this, the entire premise for the use of this model in the current study is greatly flawed.

We respectfully disagree with the reviewer's comment. While the recent study by the Lee group (Senyo et al, Nature 2013) using multi-isotope imaging mass spectrometry (MIMS) and pulsing with stable isotopes is a landmark study in the field of cardiac regeneration, it does not settle the debate regarding the origins of postnatal cardiomyogenesis. While MIMS has an unparalleled resolution capacity (of $<1\mu\text{m}^3$), it is an extremely time-consuming method that precludes analysis of large amounts of tissue (Palacios et al, EMBO Mol Med 2013). This is a critical point, as Senyo et al's conclusions regarding the cellular origins of postnatal cardiomyogenesis in infarcted hearts were derived from a comparison of the percentages of N^{15} GFP+ vs N^{15} GFP- cardiomyocytes in a total of 16 N^{15} + myocytes. This low number (n=16) is likely too small to detect differences in the rate of ^{15}N incorporation in GFP+ and GFP- myocytes. In addition, as the reviewer points out, the results of the study by Senyo et al do not agree with earlier studies from the same group (Hsieh et al, Nat Med 2007; Loffredo et al, Cell Stem Cell 2011), as in both of these earlier studies the authors reported (apart from dilution of the GFP+ myocyte pool post-MI) increased rates of BrdU incorporation in GFP- myocytes (compared to GFP+ myocytes) post-MI (hinting towards a contribution of endogenous progenitors to the myocyte pool post-injury). Finally, several other studies employing different fate mapping models indicate that progenitor cell-mediated cardiomyocyte renewal does occur following myocardial injury (Tamura et al, Arterioscler. Thromb. Vasc. Biol. 2011; Uchida et al, Stem Cell Reports. 2013, Ellison et al, Cell 2013). Taken together, we believe that the role of endogenous progenitor cells in adult cardiomyogenesis remains unclear. Thus, we believe that our study, in which we achieved prospective genetic labeling of endogenous cardioblasts (on their way to cardiomyogenic differentiation) and were therefore able to: a) provide insight into their molecular identity, functional properties and anatomic origin, and b) identify treatment strategies and paracrine signaling pathways that could be used for therapeutic stimulation of said cardioblasts, represents a valuable contribution to the field.

Second, the author's interpretation of their studies wherein reporter gene recombination is initiated after myocardial injury is dependent upon ABSOLUTE elimination of differentiated cardiomyocytes in the digestion/isolation steps. This is not likely using the protocols used, and certainly was not documented satisfactorily.

This limitation (and the extensive care we took to overcome it) has been extensively addressed in the limitations paragraph of the discussion (page 17, line 9). However, we feel that it needs to be emphasized once more that the majority of the detected labeled cells did not express sarcomeric proteins (Fig 2D), which by default excludes the possibility that they were cardiomyocytes/cardiomyocyte remnants.

The authors have also recently shown (in EMBO Medicine) that CDC delivery promotes cardiac myocyte renewal via two mechanisms - 50% via direct differentiation of the CDCs into de novo cardiomyocytes, and 50% via proliferation of pre-existing cardiomyocytes. Given the overtly quantitative nature of that earlier study, there is no additional wiggle room to account for the additional mechanisms for CDC-mediated renewal as was proposed here.

With all due respect to the reviewer, this comment is wrong. As we state in the manuscript, several studies point towards an indirect mechanism of action underlying CDC-induced therapeutic benefits; CDCs not engraft long-term into the host myocardium and do not directly differentiate into myocytes (at least to a significant degree), but they do indirectly stimulate endogenous regenerative/repairative pathways (see among others: Malliaras et al, Circulation 2012; Malliaras et al, EMBO Mol Med, 2013; Chimenti et al, Circ Res 2010; Terrovitis et al, JACC, 2009). In our previous publication (Malliaras et al, EMBO Mol Med, 2013) we show that CDCs promote cardiac regeneration via 2 mechanisms: ~50% via proliferation of pre-existing cardiomyocytes and ~50% via cardiomyogenic differentiation of *endogenous* progenitors. Here we simply attempt to characterize the cells responsible for the second component (cardiomyogenic differentiation of endogenous progenitors).

There are numerous other concerns with the work. Why do the GFP+ CPC-derived cells not survive in culture?

The inability of GFP+ cardioblasts to survive in vitro should most likely be attributed to the fact that we did not identify/develop the necessary conditions that would support their maintenance or expansion in culture. Off note, in vitro maintenance or expansion of hematopoietic stem cells (the prototypical stem cells) was considered to be very challenging for decades (Mikkola et al, Development, 2006). Furthermore, despite three decades of effort, no one has yet succeeded in long-term culture of non-transformed cardiomyocytes.

How can the authors rule out that the rare beating cell reported is not simply carryover of a small mono-nucleated cardiomyocyte?

While we cannot 100% rule out that that the observed beating cells may be small mono-nucleated myocytes, this is rather unlikely due to: a) their small size, which is >10 fold smaller compared to adult myocytes, and b) their morphology, which is round and not rod-shaped. As stated above, this limitation (and the steps we took to overcome it) is extensively discussed in the discussion of the manuscript (page 17, line 9).

How can the authors rule out that the very rare highly differentiated cardiomyocytes following engraftment of CPCs did not arise from fusion events?

We thank the reviewer for this comment. We cannot definitely rule out that the GFP+ myocytes arise from fusion events. However, it needs to be pointed out that such fusion events are exceedingly rare in the myocardium (0.014-0.27 fused cardiomyocytes/section [Alvarez-Dolado et al, Nature, 2003]); their frequency is 10-100 fold lower than the frequency of GFP+ cardiomyocytes observed in our study post-cardioblast injection (3.4-4.2 GFP+ cardiomyocytes/ section). Nevertheless, in the revised manuscript we address the possibility that GFP+ cardiomyocytes arise from cellular fusion.

On page 9, line 8 we now state: "*Rare events of cardiomyogenic differentiation could be detected in four of six cardioblast-injected hearts (2/3 normal [4.2±2.2 GFP+ cardiomyocytes/ heart section] and 2/3 infarcted [3.4±3.6 GFP+ cardiomyocytes/ heart section] hearts; Fig 5B-D). GFP+ cardiomyocytes appeared fully mature and structurally integrated with the surrounding myocardium, as they shared gap junctions with neighboring GFP- myocytes (Fig 5B). While we cannot exclude cell fusion (between injected GFP+ cells and host myocytes) as a source of GFP+*

cardiomyocytes, such fusion events are exceedingly rare in the myocardium (Alvarez-Dolado et al, 2003), occurring at a frequency 10-100 fold smaller compared to the frequency of GFP+ cardiomyocytes observed post-injection of GFP+ cardioblasts here.”

The data using BMT to demonstrate that the marrow does not carry cells with cardiomyogenic activity has been previously shown using the same model by Lee's group.

We assume that the reviewer is referring to the study by Loffredo et al (Cell Stem Cell, 2011). In that study, the authors did not perform any bone marrow transplant experiments. Instead, they isolated either c-kit cells or mesenchymal stromal cells (MSCs) from the bone marrow and delivered these cell populations intramyocardially post-MI. The authors concluded that neither bone marrow-derived c-kit cells nor bone marrow-derived MSCs transdifferentiate into cardiomyocytes after intramyocardial injection. This conclusion is considerably different from our findings in 2 ways: a) the study by Loffredo et al. focuses on purified bone marrow derived c-kit cells and MSCs (2 cell populations that comprise only a very small percentage of bone marrow cells), while we investigate all bone marrow-derived cells (given that we performed bone marrow transplants); and b) the study by Loffredo et al. investigates the cardiomyogenic potential of exogenous bone marrow-derived c-kit cells/MSCs after intramyocardial injection into the heart post-MI, while our study investigates the ability of endogenous bone marrow-derived cells to give rise to endogenous cardioblasts that are activated post-MI.

2nd Editorial Decision

27 March 2014

Please find enclosed the final report on your manuscript. We are pleased to inform you that your manuscript is accepted for publication and is soon being sent to our publisher to be included in the next available issue of EMBO Molecular Medicine.

Congratulations on your interesting work,

***** Reviewer's comments *****

Referee #1 (Comments on Novelty/Model System):

It is an interesting study to add more knowledge onto the controversial existence of cardiac progenitor cells.

Referee #1 (Remarks):

I would like to thank the authors for addressing all of my previous comments. I have no further comments on this manuscript.